# Taking Notes on the Fly Helps Language Pre-Training

**Qiyu Wu**[1][*], **Chen Xing**[2][*], **Yatao Li**[3], **Guolin Ke**[3], **Di He**[3][†], **Tie-Yan Liu**[3]
[1]Peking University
[2]College of Compute Science, Nankai University
[3]Microsoft Research
`qiyu.wu@pku.edu.cn`
`xingchen1113@gmail.com`
`{yatli, guolin.ke, dihe, tyliu}@microsoft.com`

## ABSTRACT

How to make unsupervised language pre-training more efficient and less resource-intensive is an important research direction in NLP. In this paper, we focus on improving the efficiency of language pre-training methods through providing better data utilization. It is well-known that in language data corpus, words follow a heavy-tail distribution. A large proportion of words appear only very few times and the embeddings of rare words are usually poorly optimized. We argue that such embeddings carry inadequate semantic signals, which could make the data utilization inefficient and slow down the pre-training of the entire model. To mitigate this problem, we propose Taking Notes on the Fly (TNF), which takes notes for rare words on the fly during pre-training to help the model understand them when they occur next time. Specifically, TNF maintains a note dictionary and saves a rare word's contextual information in it as notes when the rare word occurs in a sentence. When the same rare word occurs again during training, the note information saved beforehand can be employed to enhance the semantics of the current sentence. By doing so, TNF provides better data utilization since cross-sentence information is employed to cover the inadequate semantics caused by rare words in the sentences. We implement TNF on both BERT and ELECTRA to check its efficiency and effectiveness. Experimental results show that TNF's training time is 60% less than its backbone pre-training models when reaching the same performance. When trained with the same number of iterations, TNF outperforms its backbone methods on most of downstream tasks and the average GLUE score. Source code is attached in the supplementary material.

## 1 INTRODUCTION

Unsupervised language pre-training, e.g., BERT (Devlin et al., 2018), is shown to be a successful way to improve the performance of various NLP downstream tasks. However, as the pre-training task requires no human labeling effort, a massive scale of training corpus from the Web can be used to train models with billions of parameters (Raffel et al., 2019), making the pre-training computationally expensive. As an illustration, training a BERT-base model on Wikipedia corpus requires more than five days on 16 NVIDIA Tesla V100 GPUs. Therefore, how to make language pre-training more efficient and less resource-intensive, has become an important research direction in the field (Strubell et al., 2019).

Our work aims at improving the efficiency of language pre-training methods. In particular, we study how to speed up pre-training through better data utilization. It is well-known that in a natural language data corpus, words follow a heavy-tail distribution (Larson, 2010). A large proportion of words appear only very few times and the embeddings of those (rare) words are usually poorly optimized and noisy (Bahdanau et al., 2017; Gong et al., 2018; Khassanov et al., 2019; Schick & Schütze, 2020).

---

[*]Equal Contribution. Work done during internships at Microsoft Research Asia.
[†]Correspondence to:dihe@microsoft.com

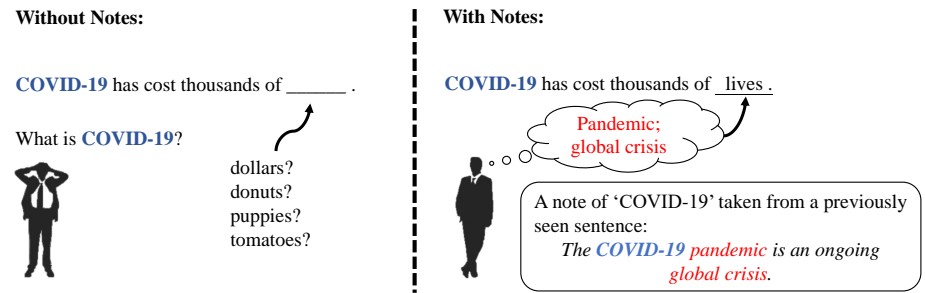

Figure 1: An illustration of how taking notes of rare words can help language understanding. The left part of the figure shows that without any understanding of the rare word "COVID-19", there are too many grammatically-correct, while semantically-wrong options for us to fill in the blank. In the right half, we show that a note of "COVID-19" taken from a previously-seen sentence can act as a very strong signal for us to predict the correct word at the masked position.

Unlike previous works that sought to merely improve the embedding quality of rare words, we argue that the existence of rare words could also slow down the training process of other model parameters. Taking BERT as an example, if we imagine the model encounters the following masked sentence during pre-training:

*COVID-19 has cost thousands of lives.*

Note that "COVID-19" is a rare word, while also the only key information for the model to rely on to fill in the blank with the correct answer "lives". As the embedding of the rare word "COVID-19" is poorly trained, the Transformer lacks concrete input signal to predict "lives". Furthermore, with noisy inputs, the model needs to take longer time to converge and sometimes even cannot generalize well (Zhang et al., 2016). Empirically, we observe that around 20% of the sentences in the corpus contain at least one rare word. Moreover, since most pre-training methods concatenate adjacent multiple sentences to form one input sample, empirically we find that more than 90 % of input samples contain at least one rare word. The large proportion of such sentences could cause severe data utilization problem for language pre-training due to the lack of concrete semantics for sentence understanding. Therefore, learning from the masked language modeling tasks using these noisy embeddings may make the pre-training inefficient. Moreover, completely removing those sentences with rare words is not an applicable choice either since it will significantly reduce the size of the training data and hurt the final model performance.

Our method to solve this problem is inspired by how humans manage information. Note-taking is a useful skill which can help people recall information that would otherwise be lost, especially for new concepts during learning (Makany et al., 2009). If people take notes when facing a rare word that they don't know, then next time when the rare word appears, they can refer to the notes to better understand the sentence. For example, we may meet the following sentence somewhere beforehand: *The COVID-19 pandemic is an ongoing global crisis.* From the sentence, we can realize that "COVID-19" is related to "pandemic" and "global crisis" and record the connection in the notes. When facing "COVID-19" again in the masked-language-modeling task above, we can refer to the note of "COVID-19". It is easy to see that once "pandemic" and "global crisis" are connected to "COVID-19", we can understand the sentence and predict "lives" more easily, as illustrated in Figure 1. Mapped back to language pre-training, we believe for rare words, explicitly leveraging cross-sentence information is helpful to enhance semantics of the rare words in the current sentence to predict the masked tokens. Through this more efficient data utilization, the Transformer can receive better input signals which leads to more efficient training of its model parameters.

Motivated by the discussion above, we propose a new learning approach called "Taking Notes on the Fly"(TNF) to improve data utilization for language pre-training. Specifically, we maintain a note dictionary, where the keys are rare words and the values are historical contextual representations of them. In the forward pass, when a rare word $w$ appears in a sentence, we query the value of $w$ in the note dictionary and use it as a part of the input. In this way, the semantic information of $w$ saved in the note can be encoded together with other words through the model. Besides updating the model parameters, we also update the note dictionary. In particular, we define the note of $w$ in the current

sentence as the mean pooling over the contextual representations of the words nearby $w$. Then we update $w$'s value in the note dictionary by a weighted linear combination of $w$'s previous value and $w$'s note in the current sentence. TNF introduces little computational overhead at pre-training since the note dictionary is updated on the fly during the forward pass. Furthermore, different from the memory-augmented neural networks (Santoro et al., 2016; Guu et al., 2020), the note dictionary is only used to improve the training efficiency of the model parameters, while not served as a part of the model. When the pre-training is finished, we discard the note dictionary and use the trained Transformer encoder during the fine-tuning of downstream tasks, same as all previous works.

We conduct experiments using BERT and ELECTRA (Clark et al., 2019) as TNF's backbone methods. Results show that TNF significantly expedites BERT and ELECTRA, and improves their performances on downstream tasks. BERT-TNF and ELECTRA-TNF's training times are both $60\%$ less than their corresponding backbone models when reaching the same performance. When trained with the same number of iterations, BERT-TNF and ELECTRA-TNF outperform the backbone methods on both the average GLUE score and the majority of individual tasks. We also observe that even in the downstream tasks where rare words only take a neglectable proportion of the data (i.e. $0.47\%$), TNF also outperforms baseline methods with a large margin. It indicates that TNF improves the pre-training of the entire model.

## 2 RELATED WORK

**Efficient BERT pre-training.** The massive energy cost of language pre-training (Strubell et al., 2019) has become an obstacle to its further developments. There are several works aiming at reducing the energy cost of pre-training. Gong et al. (2019) observes that parameters in different layers have similar attention distribution, and propose a parameter distillation method from shallow layers to deep layers. Another notable work is ELECTRA (Clark et al., 2019), which develops a new task using one discriminator and one generator. The generator corrupts the sentence, and the discriminator is trained to predict whether each word in the corrupted sentence is replaced or not. Orthogonal to them, we focus on improving pre-training efficiency by finding ways to utilize the data corpus better. Therefore, it can be applied to all of the methods above to further boost their performances.

**Representation of rare words.** It is widely acknowledged that the quality of rare words' embeddings is significantly worse than that of popular words. Gao et al. (2019) provides a theoretical understanding of this problem, which illustrates that the problem lies in the sparse (and inaccurate) stochastic optimization of neural networks. Several works attempt to improve the representation of rare words using linguistic priors (Luong et al., 2013; El-Kishky et al., 2019; Kim et al., 2016; Santos & Zadrozny, 2014). But the improved embedding quality is still far behind that of popular words (Gong et al., 2018). Sennrich et al. (2015) develops a novel way to split each word into sub-word units. However, the embeddings of low-frequency sub-word units are still difficult to train (Ott et al., 2018). Due to the poor quality of rare word representations, the pre-training model built on top of it suffers from noisy input semantic signals which lead to inefficient training. We try to bypass the problem of poor rare word representations by leveraging cross-sentence information to enhance input semantic signals of the current sentence for better model training.

**Memory-augmented BERT.** Another line of work close to ours uses memory-augmented neural networks in language-related tasks. Févry et al. (2020) and Guu et al. (2020) define the memory buffer as an external knowledge base of entities for better open domain question answering tasks. Khandelwal et al. (2019) constructs the memory for every test context at inference, to hold extra token candidates for better language modeling. Similar to other memory-augmented neural networks, the memory buffer in these works is a model component that will be used during inference. Although sharing general methodological concepts with these works, the goal and details of our method are different from them. Especially, our note dictionary is only maintained in pre-training for efficient data utilization. At fine-tuning, we ditch the note dictionary, hence adding no extra time or space complexity to the backbone models.

## 3 TAKING NOTES ON THE FLY

### 3.1 PRELIMINARY

In this section, we use the BERT model as an example to introduce the basics of the model architecture and training objective of language pre-training. BERT (**B**idirectional **E**ncoder **R**epresentation from

**T**ransformers) is developed on a multi-layer bidirectional Transformer encoder, which takes a sequence of word semantic information (token embeddings) and order information (positional embeddings) as input, and outputs the contextual representations of words.

Each Transformer layer is formed by a self-attention sub-layer and a position-wise feed-forward sub-layer, with a residual connection (He et al., 2016) and layer normalization (Ba et al., 2016) applied after every sub-layer. The self-attention sub-layer is referred to as "Scaled Dot-Product Attention" in Vaswani et al. (2017), which produces its output by calculating the scaled dot products of *queries* and *keys* as the coefficients of the *values*, i.e.,

$$\text{Attention}(Q, K, V) = \text{Softmax}(\frac{QK^T}{\sqrt{d}})V. \tag{1}$$

$Q$ (Query), $K$ (Key), $V$ (Value) are the hidden representations outputted from the previous layer and $d$ is the dimension of the hidden representations. Transformer also extends the aforementioned self-attention layer to a multi-head version in order to jointly attend to information from different representation subspaces. The multi-head self-attention sub-layer works as follows,

$$\text{Multi-head}(Q, K, V) = \text{Concat}(\text{head}_1, \cdots, \text{head}_H)W^O \tag{2}$$

$$\text{head}_k = \text{Attention}(QW_k^Q, KW_k^K, VW_k^V), \tag{3}$$

where $W_k^Q \in \mathbb{R}^{d \times d_K}, W_k^K \in \mathbb{R}^{d \times d_K}, W_k^V \in \mathbb{R}^{d \times d_V}$ are projection matrices. $H$ is the number of heads. $d_K$ and $d_V$ are the dimensions of the key and value separately.

Following the self-attention sub-layer, there is a position-wise feed-forward (FFN) sub-layer, which is a fully connected network applied to every position identically and separately. The FFN sub-layer is usually a two-layer feed-forward network with a ReLU activation function in between. Given vectors $\{h_1, \ldots, h_n\}$, a position-wise FFN sub-layer transforms each $h_i$ as $\text{FFN}(h_i) = \sigma(h_i W_1 + b_1)W_2 + b_2$, where $W_1, W_2, b_1$ and $b_2$ are parameters.

BERT uses the Transformer model as its backbone neural network architecture and trains the model parameters with the masked language model task on large text corpora. In the masked language model task, given a sampled sentence from the corpora, $15\%$ of the positions in the sentence are randomly selected. The selected positions will be either replaced by special token `[MASK]`, replaced by randomly picked tokens or remain the same. The objective of BERT pre-training is to predict words at the masked positions correctly given the masked sentences. As this task requires no human labeling effort, large scale data corpus is usually used to train the model. Empirically, the trained model, served as a good initialization, significantly improves the performance of downstream tasks.

### 3.2 TRAINING BERT BY TAKING NOTES ON THE FLY

As presented in many previous works, the poorly-updated embeddings of rare words usually lack adequate semantic information. This could cause data utilization problem given the lack of necessary semantic input for sentence understanding, thus making the pre-training inefficient. In this section, we propose a method called Taking Notes on the Fly (TNF) to mitigate this problem. For ease of understanding, we describe TNF on top of the BERT model. While TNF can be easily applied to any language pre-training methods, such as ELECTRA. The main component of TNF is a note dictionary, saving historical context representations (notes) of rare words on the fly during pre-training. In the following of the section, we introduce TNF by illustrating in detail how we construct, maintain and leverage the note dictionary for pre-training.

**The Construction of Note Dictionary.** To enrich the semantic information of rare words for a better understanding of the sentence, we explicitly leverage cross-sentence signals for those words. We first initialize a note dictionary, $\text{NoteDict}$, from the data corpus, which will maintain a note representation (value) for each rare word (key) during pre-training. Since we target at rare words, the words in the dictionary are of low frequency. However, the frequency of the words in the dictionary should not be extremely low either. It is because if the word appears only once in the corpus, there will be no "cross-sentence signal" to use. Additionally, the note dictionary also shouldn't take too many memories in practice. With all these factors taken into consideration, we define keys as those words with occurrences between $100$ and $500$ in the data corpus. The data corpus roughly contains $3.47B$ words in total and the size of $\text{NoteDict}$'s vocabulary is about $200k$.

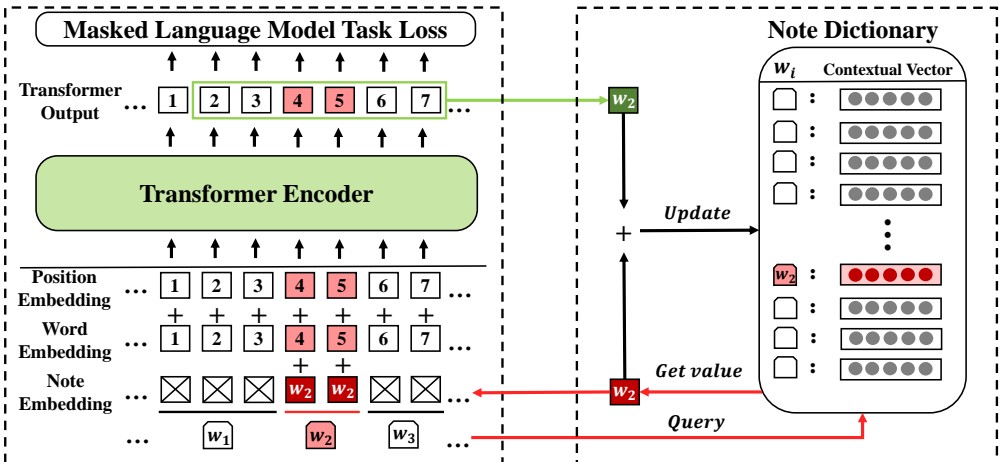

Figure 2: The training framework of Taking Notes on the FLY (TNF). The left box shows the forward pass with the help of the note dictionary. In the input word sequence, $w_2$ is a rare word. Then for tokens 4 and 5 originated from $w_2$, we query the value of $w_2$ in the note dictionary and weighted average it with token/position embeddings. The right box demonstrates how we maintain the note dictionary. After the forward pass of the model, we can get the contextual representations of the word near $w_2$ and use mean pooling over those representations as the note of $w_2$ in the current sentence. Then, we update $w_2$'s value in the note dictionary by a weighted average of the current note and its previous value.

**Maintaining Note Dictionary.** When we meet a rare word in a training sentence, we record the contextual information of its surrounding words in the sentence as its note. In detail, given a training sentence, each word will be first pre-processed into sub-word units following standard pre-processing strategies (Sennrich et al., 2015). Therefore, given a processed sequence of sub-word units (tokens), a rare word can occupy a contiguous span of tokens. For a rare word $w$ that appears both in the input token sequence $x = \{x_1, \cdots, x_i, \cdots, x_n\}$ and $\text{NoteDict}$, we denote the span boundary of $w$ in $x$ as $(s, t)$, where $s$ and $t$ are the starting and ending position. We define the note of $w$ for $x$ as

$$\text{Note}(w, x) = \frac{1}{2k + t - s} \sum_{j=s-k}^{t+k} \mathbf{c}_j, \tag{4}$$

where each $\mathbf{c}_j \in \mathbb{R}^d$ is the output of the Transformer encoder on position $j$ and served as the contextual representation of $x_j$. $k$ is half of the window size that controls how many surrounding tokens we want to take as notes and save their semantics . If we refer to the example in the introduction, the contextual representations of "pandemic" and "global crisis" are summarized in the note of "COVID-19". Note that the calculation of $\text{Note}(w, x)$ is on the fly as we can obtain $\text{Note}(w, x)$ during the forward pass using the current model. Therefore, there is no additional computational cost.

With the $\text{Note}(w, x)$ calculated with Equation 4 for the current sentence $x$, we can now update $w$'s note saved in $\text{NoteDict}$ to include the latest semantics in sentence $x$. In particular, we updates $w$'s value in $\text{NoteDict}$ using exponential moving average[1]. In this way, at any occurrence of $w$ during pre-training, its contextual information from all previous occurrences can be leveraged and used.

$$\text{NoteDict}(w) = (1 - \gamma) \cdot \text{NoteDict}(w) + \gamma \cdot \text{Note}(w, x), \tag{5}$$

where $\gamma \in (0, 1)$ is the discount factor.

**Leveraging Note Dictionary for Pre-training.** $\text{NoteDict}$ explicitly contains surrounding contexts for rare words. We use such information as a part of the input to the Transformer encoder. For any masked token sequence $x = \{x_1, \cdots, x_i, \cdots, x_n\}$, we first find all rare words that appears in both $\text{NoteDict}$ and $x$. Assume there are $m$ rare words satisfying the conditions, denoted as

---

[1]All values in $\text{NoteDict}$ are randomly initialized using the same way as word/positional embeddings.

$\{(w_j, s_j, t_j)\}_{j=1}^m$ where $s_j$ and $t_j$ are the boundary of $w_j$ in $x$. At the $i$-th position, the input to the model is defined as

$$\text{input}_i = \begin{cases} (1-\lambda) \cdot (\text{pos\_emb}_i + \text{token\_emb}_i) + \lambda \cdot \text{NoteDict}(w_j) & \exists j, s.t. \ s_j < i < t_j, \\ \text{pos\_emb}_i + \text{word\_emb}_i & \text{otherwise.} \end{cases} \tag{6}$$

$\lambda$ is a hyper-parameter controlling the degree to which TNF relies on historical context representations (notes) for rare words. We empirically set it as $0.5$.

In the standard Transformer model, at position $i$, the input to the first Transformer layer is the sum of the positional embedding $\text{pos\_emb}_i$ and the token embedding $\text{token\_emb}_i$. In Equation 6, when the token $x_i$ is originated from a rare word $w_j$ in NoteDict, we first query $w_j$ in NoteDict and then weight-averages its value $\text{NoteDict}(w_j)$ with the token embedding $\text{token\_emb}_i$ and positional embedding $\text{pos\_emb}_i$. In such a way, the historical contextual information of rare word $w_j$ in $\text{NoteDict}(w_j)$, can be processed together with other words in the current sentence in the stacked Transformer layers, which can help the model to better understand the input sequence. Figure 2 gives a general illustration of TNF in pre-training.

**Fine-tuning.** Our goal is to make the training of the model (e.g., the parameters in the Transformer encoder) more efficient. To achieve this, we leverage cross-sentence signals of rare words as notes to enrich the input signals. To verify whether the Transformer encoder is better trained with TNF, we purposely remove the NoteDict for fine-tuning and only use the trained encoder in the downstream tasks. First, in such a setting, our method can be fairly compared with previous works and backbone models, as the fine-tuning processes of all the methods are exactly the same. Second, by doing so, our model occupies no additional space in deployment, which is an advantage compared with existing memory-augmented neural networks (Santoro et al., 2016; Guu et al., 2020). We also conduct an ablation study on whether to use NoteDict during fine-tuning. Details can be found in Section 4.

## 4 EXPERIMENTS

To verify the efficiency and effectiveness of TNF, we conduct experiments and evaluate pre-trained models on fine-tuning downstream tasks. All codes are implemented based on *fairseq* (Ott et al., 2019) in *PyTorch* (Paszke et al., 2017). All models are run on 16 NVIDIA Tesla V100 GPUs with mixed-precision (Micikevicius et al., 2017).

### 4.1 EXPERIMENTAL SETUP

To show the wide adaptability of TNF, we use BERT (Devlin et al., 2018) and ELECTRA (Clark et al., 2019) as the backbone language pre-training methods and implement TNF on top of them. We fine-tune the pre-trained models on GLUE (**G**eneral **L**anguage **U**nderstanding **E**valuation) (Wang et al., 2018) to evaluate the performance of the pre-trained models. We follow previous work to use eight tasks in GLUE, including CoLA, RTE, MRPC, STS, SST, QNLI, QQP, and MNLI.The detailed setting of the fine-tuning is illustrated in Appendix A.1.

**Data Corpus and Pre-training Tasks.** Following BERT (Devlin et al., 2018), we use the English Wikipedia corpus and BookCorpus (Zhu et al., 2015) for pre-training. By concatenating these two datasets, we obtain a corpus with roughly 16GB in size, similar to Devlin et al. (2018). We also follow a couple of consecutive pre-processing steps: segmenting documents into sentences by Spacy[2], normalizing, lower-casing, tokenizing the texts by Moses decoder (Koehn et al., 2007), and finally, applying byte pair encoding (BPE) (Sennrich et al., 2015) with the vocabulary size set as 32,678. We use *masked language modeling* as the objective of BERT pre-training and *replaced token detection* for ELECTRA pre-training. We remove the next sentence prediction task and use *FULL-SENTENCES* mode to pack sentences as suggested in RoBERTa (Liu et al., 2019). Details of the two pre-training tasks and TNF's detailed implementation on ELECTRA can be found in Appendix A.2

---

[2]`https://spacy.io`

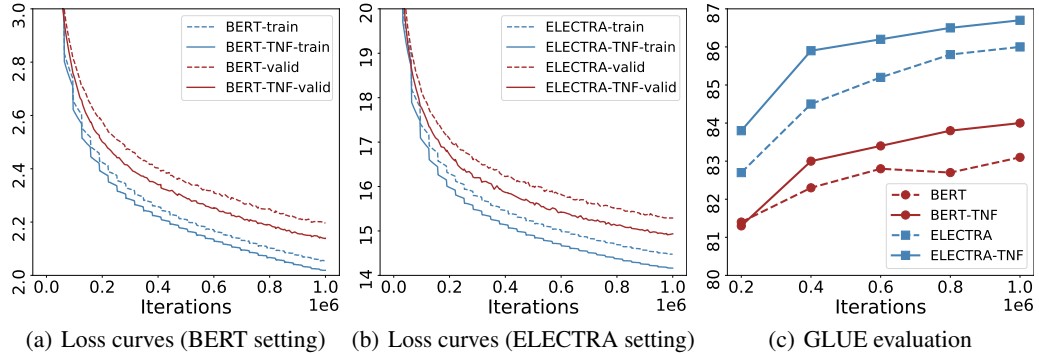

(a) Loss curves (BERT setting)  (b) Loss curves (ELECTRA setting)  (c) GLUE evaluation

Figure 3: The curves of pre-training loss, pre-training validation loss and average GLUE score for all models trained under the BERT setting and ELECTRA setting. All three sub-figures show that TNF expedites the backbone methods.

**Model architecture and hyper-parameters.**   We conduct experiments on BERT (110M parameters) (Devlin et al., 2018) and ELECTRA (110M parameters) (Clark et al., 2019) (i.e., the base setting). A 12-layer Transformer is used for BERT. For each layer, the hidden size is set to 768 and the number of attention head ($H$) is set to 12. ELECTRA composes of a discriminator and a generator. The discriminator is the same as BERT and the generator is $1/3$-width BERT model, suggested by the original paper (Clark et al., 2019). We also conduct experiments on large models (335M parameters), details are at Appendix A.5. We use the same pre-training hyper-parameters for all experiments. All models are pre-trained for $1000k$ steps with batch size 256 and maximum sequence length 512. All hyper-parameter configurations are reported in Appendix A.3.

## 4.2 RESULTS AND ANALYSIS

**TNF improves pre-training efficiency.**   Figure 3 shows for all pre-training methods, how the pre-training loss, pre-training validation loss and average GLUE score change as pre-training proceeds. From Figure 3(a) and (b), we can see that as the training proceeds, TNF's pre-training loss and validation loss is constantly lower than its corresponding backbone methods. It indicates that TNF has accelerated its backbone model through the entire pre-training process. We can also notice from Figure 3(a) and (b) that the gap between the losses of the backbone model and TNF keeps increasing during pre-training. A possible explanation of this phenomenon is that the qualities of notes would improve with pre-training. Therefore, the notes that TNF takes for rare words could contain better semantic information to help the encoder as the training goes.

From Figure 3(c), we can see that the average GLUE score of TNF is also larger than the baseline through most of the pre-training. TNF's GLUE scores at 400k iteration are competitive to those of the corresponding backbone models at 1000k

|  | Params | Avg. GLUE |
|---|---|---|
| GPT-2 | 117 M | 78.8 |
| BERT | 110 M | 82.2 |
| SpanBERT | 110 M | 83.9 |
| ELECTRA | 110 M | 85.1 |
| BERT (Ours) | 110 M | 83.1 |
| BERT-TNF | 110 M | 83.9 |
| ELECTRA (Ours) | 110 M | 86.0 |
| **ELECTRA-TNF** | 110 M | **86.7** |

Table 1: Average GLUE score of all methods on the dev set when the pre-training finished, i.e., at $1e6$ iterations. Results of GPT, BERT and ELECTRA are from Clark et al. (2019). The result of SpanBERT is obtained by fine-tuning the released checkpoint from Joshi et al. (2019). We also reproduce BERT and ELECTRA in our system for fair comparison. We report their results as BERT(ours) and ELECTRA(ours).

iteration in both BERT and ELECTRA settings. It means that to reach the same performance, TNF can save $60\%$ of pre-training time. If models are trained on 16 NVIDIA Tesla V100 GPUs, BERT-TNF can reach BERT's final performance within 2 days while it takes BERT 5.7 days.

|  | MNLI | QNLI | QQP | SST | CoLA | MRPC | RTE | STS | Avg. |
|---|---|---|---|---|---|---|---|---|---|
| BERT (Ours) | 85.0 | **91.5** | 91.2 | 93.3 | 58.3 | 88.3 | 69.0 | 88.5 | 83.1 |
| BERT-TNF | 85.0 | 91.0 | **91.2** | 93.2 | 59.5 | **89.3** | **73.2** | **88.5** | **83.9** |
| BERT-TNF-F | **85.1** | 90.8 | 91.1 | 93.3 | 59.8 | 88.8 | 72.1 | 88.5 | 83.7 |
| BERT-TNF-U | 85.0 | 90.9 | 91.1 | **93.4** | **60.2** | 88.7 | 71.4 | 88.4 | 83.6 |
| ELECTRA(Ours) | 86.8 | 92.7 | 91.7 | 93.2 | 66.2 | **90.2** | 76.4 | **90.5** | 86.0 |
| ELECTRA-TNF | **87.0** | **92.7** | **91.8** | 93.6 | **67.0** | 90.1 | 81.2 | 90.1 | **86.7** |
| ELECTRA-TNF-F | 86.9 | 92.6 | 91.8 | **93.7** | 65.9 | 89.7 | **81.4** | 89.8 | 86.5 |
| ELECTRA-TNF-U | 86.9 | 92.7 | 91.7 | 93.6 | 66.3 | 89.8 | 81.0 | 89.8 | 86.5 |

Table 2: Performance of different models on downstream tasks. Results show that TNF outperforms backbone methods on the majority of individual tasks. We also list the performance of two variants of TNF. Both of them leverage the node dictionary during fine-tuning. Specifically, TNF-F uses fixed note dictionary and TNF-U updates the note dictionary as in pre-training. Both models outperforms the baseline model while perform slightly worse than TNF.

Beyond the base-sized models (110 M parameters), we also apply TNF on large models to check the effectiveness of our method. Details are reported at Appendix A.5.

**TNF improves its backbone model's performance.** BERT models are severely under-trained (Liu et al., 2019). Therefore, training faster usually indicates better final performance given the same amount of pre-training time. In Table 1, we present the average GLUE score of all methods when the pre-training finished, i.e., at $1M$ updates. We can see from the table that in both BERT and ELECTRA settings, TNF outperforms its backbone methods on the average GLUE score with a large margin. Among them, ELECTRA-TNF's performance outperforms all state-of-the-art baseline methods with a similar model size. In Table 2, we present the performance of TNF and its backbone methods on GLUE sub-tasks. TNF outperforms its backbone models on the majority of sub-tasks. TNF's performance improvement against the baseline is most prominent on sub-tasks with smaller datasets. Among all 8 sub-tasks, RTE has the smallest training set which contains 2.5k training samples in total (Wang et al., 2018). On RTE, TNF obtains the biggest performance improvement (4.2 and 4.8 points for BERT and ELECTRA, respectively) compared with the baseline. On another small-data sub-tasks CoLA, TNF also outperforms the baseline with considerable margins (1.2 and 0.8 points for BERT and ELECTRA respectively). This indicates that TNF pre-training can indeed provide a better initialization point for fine-tuning, especially on downstream tasks with smaller data sizes.

**Empirical analysis on whether to keep notes during fine-tuning.** As mentioned in Section 3, when fine-tuning the pre-trained models on downstream tasks, TNF doesn't use the note dictionary. One may wonder what the downstream task performance would be like if we keep the note dictionary in fine-tuning. To check this, we test two TNF's variations for comparison. The first variation is denoted as TNF-F, in which we fix the noted dictionary and use it in the forward pass during fine-tuning as described in Equation 6. The second variation is denoted as TNF-U. In TNF-U, we not only use the note dictionary, but also add the note dictionary into the computation graph and update the note representations by back-propagation. The results are listed in Table 2. The results show that both TNF-F and TNF-U outperform the backbone model. This indicates that no matter if we keep the notes at fine-tuning or not, TNF can boost its backbone pre-training method's performance. Moreover, we also observe that their performances are both slightly worse than TNF. We hypothesize the reason behind can be the distribution discrepancy of the pre-training and fine-tuning data. More detailed analysis can be found in Appendix A.4.

To see how pre-training with notes affects the model performance, we further study the validation loss at the pre-training stage in different settings. We firstly study the validation MLM loss on sentences without rare words on both BERT and BERT-TNF. We find that at iteration 200k, BERT's MLM loss on sentences without rare words is 3.896. While BERT-TNF's MLM loss on sentences without rare words is 3.869, less than that of BERT. This indicates that with TNF, the model is in general better trained to preserve semantics related to common context. Then we calculate the validation loss on sentences with rare words for three model settings, a pre-trained TNF model with/without using the notes and a standard pre-trained BERT. We find that the loss order is TNF with notes < BERT <

TNF without notes. This indicates that information related to rare words are contained in the TNF notes but not memorized in the Transformer parameters.

Furthermore, we conduct sensitivity analysis of the newly-added hyper-parameters of TNF. Details and complete results can be found at Appendix A.4.

## 5 CONCLUSION

In this paper, we focus on improving the data utilization for more efficient language pre-training through the lens of the word frequency. We argue the large proportion of rare words and their poorly-updated word embeddings could slow down the entire pre-training process. Towards this end, we propose Taking Notes on the Fly (TNF). TNF alleviates the heavy-tail word distribution problem by taking temporary notes for rare words during pre-training. In TNF, we maintain a note dictionary to save historical contextual information for rare words when we meet them in training sentences. In this way, when rare words appear again, we can leverage the cross-sentence signals saved in their notes to enhance semantics to help pre-training. TNF saves $60\%$ of training time for its backbone methods when reaching the same performance. If trained with the same number of updates, TNF outperforms backbone pre-training methods by a large margin in downstream tasks.

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

# A  APPENDIX

## A.1  GLUE FINE-TUNING.

We fine-tune the pre-trained models on GLUE (General Language Understanding Evaluation) (Wang et al., 2018) to evaluate the performance of the pre-trained models. We follow previous work to use eight tasks in GLUE, including CoLA, RTE, MRPC, STS, SST, QNLI, QQP, and MNLI. For evaluation metrics, we report Matthews correlation for CoLA, Pearson correlation for STS-B, and accuracy for other tasks. We use the same optimizer (Adam) with the same hyper-parameters as in pre-training. Following previous work, we search the learning rates during the fine-tuning for each downstream task. The details are listed in Table 3. For fair comparison, we do not apply any published tricks for fine-tuning. Each configuration is run five times with different random seeds, and the *average* of these five results on the validation set is calculated as the final performance of one configuration. We report the best number over all configurations for each task.

## A.2  PRE-TRAINING TASKS.

We use masked language modeling as the objective of BERT-based pre-training and replaced token detection for ELECTRA-based pre-training. We remove the next sentence prediction task and use *FULL-SENTENCES* mode to pack sentences as suggested in RoBERTa (Liu et al., 2019).

**Masked Language Modeling of BERT.**   The masked probability is set to $0.15$ with *whole word masking*. As mentioned above, our datasets are processed as sub-word tokens after BPE. *Whole word masking* here means when a sub-word is masked, we also mask all surrounding tokens that originate from the same word. After masking, we replace 80% of the masked positions with `[MASK]`, 10% by randomly sampled words, and keep the remaining 10% unchanged.

**Replaced Token Detection of ELECTRA.**   We use the output of the generator of ELECTRA to calculate note representations and update the note dictionary. Then we only apply the notes on the input of the discriminator (i.e., adding the note representations of rare words together with the token embeddings as the input of the discriminator), not on the input of the generator. The reason is that as shown in BERT-TNF's experiments, the notes can enhance the training of the generator. However, an overly strong generator may pose an unnecessarily challenging task for the discriminator (Clark et al., 2019), leading to unsatisfactory pre-training of the discriminator. The masked probability of the generator is set to $0.15$ with whole word masking and all masked positions are replaced with `[MASK]`.

## A.3  HYPER-PARAMETERS

We conduct experiments on BERT-Base (110M parameters), BERT-Large (335M parameters) (Devlin et al., 2018) and ELECTRA (Clark et al., 2019). BERT consists of 12 and 24 Transformer layers for the base and large model, respectively. For each layer, the hidden size is set to 768 and 1024 and the number of attention head ($H$) is set to 12 and 16 for the base and large model. The architecture of the discriminator of ELECTRA is the same as BERT-Base. The size of the generator is $1/3$ of the discriminator. We use the same pre-training hyper-parameters for all experiments. All models are pre-trained for $1000k$ steps with batch size 256 and maximum sequence length 512. We use Adam (Kingma & Ba, 2014) as the optimizer, and set its hyperparameter $\epsilon$ to 1e-6 and $(\beta 1, \beta 2)$ to (0.9, 0.98). The peak learning rate is set to 1e-4 with a $10k$-step warm-up stage. After the warm-up stage, the learning rate decays linearly to zero. We set the dropout probability to 0.1 and weight decay to 0.01. There are three additional hyper-parameters for TNF, half window size $k$, discount factor $\lambda$ and weight $\gamma$. We set $k$ as 16, $\lambda$ as 0.5, $\gamma$ as 0.1 for the main experiment, except for ELECTRA $k$ is empirically set as 32. All hyper-parameter configurations are reported in Table 3.

## A.4  ABLATION STUDY AND PARAMETER SENSITIVITY

**Empirical analysis on whether to keep notes during fine-tuning.**   We test two TNF's variations for comparison. The first variation is denoted as TNF-F, in which we fix the note dictionary and use it in the forward pass during fine-tuning as described in Equation 6. The second variation is denoted

|  | Pre-training | Fine-tuning |
|---|---|---|
| **Max Steps** | $1M$ | - |
| **Max Epochs** | - | 5 or 10 |
| **Learning Rate** | 1e-4 | {1e-5, 2e-5, 3e-5, 4e-5, 5e-5} |
| **Batch Size** | 256 | 32 |
| **Warm-up Ratio** | 0.01 | 0.06 |
| **Sequence Length** | 512 | 512 |
| **Learning Rate Decay** | Linear | Linear |
| **Adam** $\epsilon$ | 1e-6 | 1e-6 |
| **Adam** $(\beta_1, \beta_2)$ | (0.9, 0.98) | (0.9, 0.98) |
| **Dropout** | 0.1 | 0.1 |
| **Weight Decay** | 0.01 | 0.01 |
| $k$ **of BERT-TNF** | 16 | - |
| $\lambda$ **of BERT-TNF** | 0.5 | - |
| $\gamma$ **of BERT-TNF** | 0.1 | - |
| $k$ **of ELECTRA-TNF** | 32 | - |
| $\lambda$ **of ELECTRA-TNF** | 0.5 | - |
| $\gamma$ **of ELECTRA-TNF** | 0.1 | - |

Table 3: Hyper-parameters for the pre-training and fine-tuning on all language pre-training methods, include both backbone methods and TNFs.

|  | Effect of varying $k$ | | | | Effect of varying $\lambda$ | | | | |
|---|---|---|---|---|---|---|---|---|---|
| Run # | R1 | R2 | R3 | R4 | R5 | R6 | R7 | R8 | R9 |
| $k$ | 4 | 8 | 16 | 32 | 16 | 16 | 16 | 16 | 16 |
| $\lambda$ | 0.5 | 0.5 | 0.5 | 0.5 | 0.1 | 0.3 | 0.5 | 0.7 | 0.9 |
| $\gamma$ | 0.1 | 0.1 | 0.1 | 0.1 | 0.1 | 0.1 | 0.1 | 0.1 | 0.1 |
| Model Size | base | base | base | base | base | base | base | base | base |
| Avg. GLUE | 82.5 | 83.3 | 83.9 | 83.5 | 83.3 | 83.9 | 83.9 | 82.8 | 83.8 |

|  | Effect of model size | | | | Effect of varying $\gamma$ | | | | |
|---|---|---|---|---|---|---|---|---|---|
| Run # | R10 | R11 | R12 | R13 | R14 | R15 | R16 | R17 | R18 |
| $k$ | - | 16 | - | 16 | 16 | 16 | 16 | 16 | 16 |
| $\lambda$ | - | 0.5 | - | 0.5 | 0.5 | 0.5 | 0.5 | 0.5 | 0.5 |
| $\gamma$ | - | 0.1 | - | 0.1 | 0.1 | 0.3 | 0.5 | 0.7 | 0.9 |
| Model Size | base | base | large | large | base | base | base | base | base |
| Avg. GLUE | 83.1 | 83.9 | 84.4 | 85.6 | 83.9 | 83.1 | 82.9 | 83.5 | 83.0 |

Table 4: Experimental results on the sensitivity of BERT-TNF's hyper-parameter $k$, $\lambda$ and $\gamma$.

as TNF-U. In TNF-U, we not only use the note dictionary, but also add the note dictionary into the computation graph and update the note representations by back-propagation. As shown in Table 2, both TNF-F and TNF-U outperform the backbone models. This indicates that no matter if we keep the notes at fine-tuning or not, TNF can boost its backbone pre-training method's performance. Moreover, we also observe that their performances are both slightly worse than TNF. We hypothesize the reason behind can lie in the discrepancy of the pre-training and fine-tuning data. We notice that the proportion of rare words in downstream tasks are too small (from $0.47\%$ to $2.31\%$). When the data distribution of the pre-training data set is very different from the downstream data sets, notes of rare words in pre-training might be not very effective in fine-tuning.

**Sensitivity of hyper-parameters.** We also conduct experiments on the BERT model to check if TNF's performance is sensitive to the newly introduced hyper-parameters. Results are shown in Table 4. Overall, in most settings (R1-R9 and R14-R18) of varying $k$, $\lambda$ and $\gamma$, TNF outperforms the BERT-Base (R10), which indicates that TNF is generally robust to the new hyper-parameters. The experimental results using different $k$ (R1-R4) show that a larger $k$ usually leads to better performances. The reason may be that the note representation of rare words can contain more sufficient contextual information when a relatively large $k$ is applied. We also tried fixing $k$ and

tuning $\lambda$ (R5-R9) and $\gamma$ (R14-R18). We empirically find that with $\lambda = 0.5$ and $\gamma = 0.1$, BERT-TNF produces the best performance. We speculate that small $\lambda$ and $\gamma$ can make the training more stable.

## A.5 LARGE MODELS

In addition to the experiments on base models, we also train large models to check the effectiveness of TNF. A 24-layer Transformer is used for BERT-large. The hidden size is set to 1024 and the number of attention head is set to 16. Other settings are same as base models. Although it can be seen from the experiments of the previous works (Clark et al., 2019) that improving a larger model's performance on downstream tasks is usually more challenging, TNF can still save at least 40% training time on BERT-Large as shown in Figure 4. In Table 4 (R10-R13), we compare TNF's performance on BERT-Base and BERT-Large. TNF gives a larger improvement on the BERT-large (1.2 point) than BERT-base (0.7 point) when the pre-training is finished. It indicates that TNF is not only robust to the model size, but also more effective at improving the final performance when the model gets bigger.

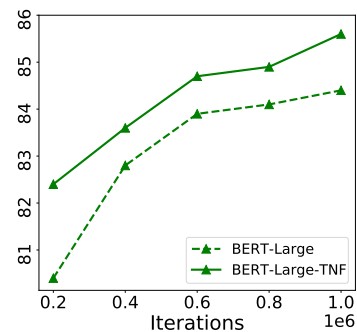

Figure 4: GLUE score of large models