# OpenReview forum: "Taking Notes on the Fly Helps Language Pre-Training"
_ICLR.cc/2021/Conference — ICLR 2021 Poster_

### Official Review · AnonReviewer3 · 2020-10-25
**Interesting paper with strong empirical results, but lacks convincing explanations for why the method works well.**

**Rating:** 7
**Confidence:** 4

**Review:**

*Summary*: This paper proposes a method for improving pretraining convergence speed by augmenting the representations of rare words with the mean-pooled representations from their previously-occuring contexts (“notes”, stored in a “note dictionary”). The method considerably speeds up the convergence of pretraining BERT and ELECTRA, and the authors furthermore show that these models perform better when fine-tuning on downstream GLUE tasks (likely because the models were undertrained to begin with, so converging faster alleviates this issue).

*Strengths*: The method is surprisingly simple and empirically quite effective. It's especially interesting to see that BERT + TNF at 400K steps has better GLUE performance than BERT at 1M steps.

*Weaknesses*: the paper does not do a convincing job of arguing that the reasons for the faster convergence comes from better modeling of rare words---I’m still not entirely sure why this works so well. Do these rare words commonly show up in GLUE (and thus, the method is helping because your representations of rare words are better)? It seems like TNF is actually improving the representations of more-common words as well.

*Recommendation*: 7 Despite the lack of clarity around why exactly this method works so well, the method seems empirically useful and straightforward to apply. I expect that this will be useful to practitioners interested in applying BERT and similar pretraining strategies to new corpora and domains.

*Questions*:

It’s a bit unclear to me that note-taking itself is required for this to work well...in the COVID example presented in the introduction, if you see the sentence “The COVID-19 pandemic is an ongoing global crisis”, isn’t it possible that MLM itself is sufficient to associate the embedding of “COVID-19” with “pandemic” and “global crisis”? Do you have further evidence to show that note-taking is actually improving the representations of rare words, besides GLUE score (which might not be very indicative, since the rare words might not show up in GLUE).

The Construction of Note Dictionary: Does 3.47B refer to the number of types or the number of tokens? Why not define keys with frequencies less than 100 in the dictionary as well (since you only use types that show up between 100 to 500 times)?

“It means that to reach the same performance, TNF can save 60% of pre-training time. If models are trained on 16 NVIDIA Tesla V100 GPUs, BERT-TNF can reach BERT’s final performance within 2 days while it takes BERT 5.7 days.”: Is the 2 days vs 5.7 days an actual wallclock measurement? Or, are you hypothesizing this based off of the loss curves?

*Missing / Erroneous Citations:*

“It is well-known that in a natural language data corpus, words follow a heavy-tail distribution (Larson, 2010)” This is more-commonly known in the NLP community as Zipf’s law. Better cites would be:
  - Zipf G. The Psychobiology of Language. London: Routledge; 1936.
  - Zipf G. Human Behavior and the Principle of Least Effort. New York: Addison-Wesley; 1949.

*Miscellaneous comments:*

“Moreover, completely removing those sentences with rare words is not an applicable choice either since it will significantly reduce the size of the training data and hurt the final model performance.”: I agree that it’s a bad idea to remove sentences with rare words, but I disagree that the issue is reducing the size of the data---you can always go collect more data and filter it to not include rare words. It’s more likely that the issue is that removing sentences with rare words would reduce the diversity of the pretraining data, which would be harmful

“Our method to solve this problem is inspired by how humans manage information.”: I think the connection to human note-taking is tenuous at best, and would omit it; the motivation remains clear without this.

---

> ### Author Response · Authors · 2020-11-17
> **Response to AnonReviewer3**
>
>
> We would like to thank Reviewer 3’s support and constructive comments. We notice that similar to Reviewer 1, Reviewer 3 also has concerns about our motivation and analysis about why TNF works. Therefore, we first describe why our method works and present empirical evidence, which we think can largely help the reviewer better understand our paper. Then we answer each question.
>
> * The motivation of TNF
>
>
> First of all, we would like to emphasize that poor rare word embeddings will hurt the training of all model parameters (such as the Transformer layers). The reasons behind are as follows. According to many previous works (see the related work section), rare words' embeddings are usually poorly optimized. There are even recent works suggesting that rare word embeddings act as noise[1]. Training a model from noisy inputs is less effective in general (see [2](Figure 1a) for more observations about how noisy inputs hurt the optimization and generalization of a machine learning model). For language pre-training tasks specifically, the input sequences with noisy rare word embeddings contain less semantically meaningful information for the Transformer to learn, making the whole model training ineffective.
>
> Given the above facts, we aim to reduce the noise from rare word embeddings in a sentence, to improve the pre-training of the whole model. Specifically, we achieve this by providing more precise semantics of rare words in a sentence using surrounding contexts from other sentences (contextualized information saved in notes). For the original method, Transformer receives thousands of noisy embeddings like 'Covid-19' containing little semantic meanings. Suppose we equip 'Covid-19' with a note that contains previous surrounding contextual information such as 'pandemic' and 'global crisis' (which are popular words). One could imagine that the sentence `Covid-19 (+pandemic + global crisis) causes thousand of lives' could have more precise semantics, which makes the training of the Transformer model more effective.
>
> We conduct additional experiments to check whether the Transformer model is better trained in BERT-TNF. To show this, we calculate MLM loss on sentences in the validation set that do not have rare words for both BERT and BERT-TNF at different checkpoints. The total number of sampled sentences that satisfy the condition is roughly 20k.  As those testing sentences don't contain rare words, the note dictionary will not be called, and we can purely compare the performance between the two Transformer models in BERT and BERT-TNF. From the table below, we can see that BERT-TNF's loss is consistently lower than BERT at all checkpoints, which suggests the entire Transformer model was improved using our method.
>
>
> #Iter   20k    50k  100k  200k   400k  600k  800k 1000k
>
> BERT 2.709 2.180 1.947 1.788 1.667 1.602 1.556 1.522
>
>  TNF  2.692 2.145 1.902 1.736 1.619 1.558 1.513 1.479
>
>
> In the submission, we also showed that TNF without notes works well on finetuning downstream tasks. Moreover, even for sub-tasks with almost no rare words occurring in the training set (0.47% rare word coverage in CoLA), BERT-TNF can still outperform BERT on it.
>
> All empirical results above support our motivation and indicate that the entire model is better pre-trained with TNF.
>
> * Regarding the frequency range of words in the note dictionary.
>
> Thanks for this question. We have tuned this frequency range to build the note dictionary before the submission. We apologize for not having put the results into the paper. Specifically, we have tried setting the range to be 50-100 and 100-1000.
>
> Range 50-100 performs almost identical to the BERT baseline. The reason is that rare words with numbers of occurrences between 50-100 can only cover 0.4% of the whole training corpora. With such low coverage, very limited training sentences would be enhanced by rare word notes. As a comparison, words whose occurrences between 100-500 cover 2% of the training corpora, which is shown empirically that could impact the pre-training. We cannot further include lower-frequency (e.g., < 50) words in the dictionary as the number of such words would exponentially increase due to the long-tail problem of language corpora, occupying a massive amount of GPU memory.  Range 100-1000 performs similar to our main results from 100-500, which suggests the rare word range in TNF is robust to some extent.
>
> The '3.4B' refers to the total number of words in the 16G training corpora.
>
> * Regarding the measurement of acceleration.
>
> Yes, the 2 days vs 5.7 days is actual wallclock measurement.
>
> ----
>
> We hope the above responses can address your questions about the design choices. Please let us know if you have further questions!
>
> [1] Li, Yangming, et al. "Handling Rare Entities for Neural Sequence Labeling.", (ACL 2020).
>
> [2] Zhang, Chiyuan, et al. "Understanding deep learning requires rethinking generalization."(ICLR 2017).

---

### Official Review · AnonReviewer4 · 2020-10-27
**Technique is simple and results are good, but too many questions remain**

**Rating:** 6
**Confidence:** 4

**Review:**

This paper proposes Taking Notes on the Fly, a technique to improve the training efficiency of language-modeling style pretraining. It works by identifying rare words in the pre-training and adding a “note-taking” component to the masked language model which augments these words with an extra “note” embedding at the input layer. The note embedding is constructed from an exponential moving average of mean-pooled contextualized representations of context windows in which that word was previously seen during training. The notes are dropped in fine-tuning. Experiments find that this pre-training method improves fine-tuning results on English NLP tasks in the GLUE benchmark when used in the original BERT pre-training setup. In particular, the model can achieve similar performance to the original BERT model with less than 40% of the training steps, and similarly for ELECTRA.

### Strengths

This paper is clearly written, the proposed technique is simple, and the results seem strong. It is laudable that the authors give experiments in the appendix to give a sense of hyperparameter sensitivity. The paper has a strong backbone and it seems that the proposed technique or something similar may serve as the basis for solid future work.

### Weaknesses

While the backbone of the paper is strong, I think it could be improved in its head (motivation) and legs (experimental studies).

First, motivation. While the framing around rare words with the COVID-19 example is interesting, I think it has gaps. The introduction argues that since “COVID-19” is a rare word, in the course of training the model may lack the necessary signal to predict the masked word “lives.” But isn’t this fact exactly what should lead the model to improve its embedding of “COVID-19”? Because gradients flow into the embeddings both through the softmax layer and the input layer.

So while adding to the context may help the model get a foothold with more effective training signal for the masked token, it seems to me that the note could also “explain away” the rare word’s embedding in the input layer, reducing the learning signal on it. If that’s the case, then to the extent that TNF works, it would be by the tradeoff between improving the learning signal at the output layer for all words (and in contextualization) and degrading it at the input layer (for rare words).

As a broader example, see https://openreview.net/pdf?id=3Aoft6NWFej. That paper argues for a masking scheme which eliminates easy shortcuts from the prediction problem to increase learning efficiency, whereas this paper argues essentially the opposite—that shortcuts must be added to hard cases in order to facilitate learning. It seems that there may be a line to walk here between a task being too hard to learn from and too easy to be useful. Because it’s not clear where that line is, I think it’s not enough to motivate TNF from only one direction. It would be better to also have an explanation of why the note-taking approach does not also make things “too easy.” It’s not obvious to me how to best make this argument, though results from some of the ablations I will suggest below might help.

This brings me to my second point: Ablation experiments. If the motivation is to improve the representations of rare words in the input, then there are even simpler ways to do this. Experiments with simple baselines and ablations are important for figuring out why exactly TNF works.

First, if the note is such a useful addition to the word embedding, why not just use it to update the embeddings directly? At that rate, the method for constructing the note embeddings looks quite similar to word embedding training objectives like word2vec and GloVe. This suggests a critical ablation:

* Initialize the word embeddings with word2vec, GloVe, or similar run over the wordpieces in the pretraining corpus. (Weirdly, I can’t find an example of this in the literature. It seems like an obvious thing to try. I may have just missed it.) Indeed, it seems to me that the framing in the paper could just as easily motivate this (much simpler) technique than TNF.

If TNF outperforms the critical ablation, that implies that its gains are coming from some of the other particulars of the technique, such as 1) the extra degree of freedom provided by decoupling the note embeddings from the wordpiece embeddings, or 2) the use of contextualized vectors for note embeddings (rather than the non-contextualized ones in the word embedding objectives).

To investigate these issues, I would suggest three more ancillary ablations on TNF:
* Directly update the rare word’s embedding with a version of Eq. 5 rather than keeping a separate note dictionary.
* Update the note embeddings via backprop instead of Eq. 5. This would amount to “partially tying” the input and output embeddings, giving more freedom to the input layer, which is partly what’s happening in TNF.
* Pool over non-contextualized instead of contextualized representations in Eq. 4.

Finally, to address the “too easy” vs “too hard” distinction, two more ablations that might help would be:
* Instead of using an exponential moving average for the note embedding update, just use the pooled context vectors from the last instance of the rare word (i.e., set $\gamma$ to 1 in Eq. 5).
* instead of using an explicit note dictionary, augment the input context with retrieved text containing the rare word. See TEK-enriched representations (https://arxiv.org/pdf/2004.12006.pdf) for an example of this. For consistency, the exact last-seen context of the rare word could be used.

The first will help identify to what extent aggregating over many multiple inputs to get a high quality representation is necessary for TNF. This could then serve as a reference point for the second ablation, which may help determine whether the fixed embedding size and pooling operation helps by creating a bottleneck for the retrieved information and preventing things from getting “too easy.” (although context window sizes might also be a confound here, that could also be controlled carefully.)

All together I think these ablations would shed a lot of light on why TNF works, and make this work much more useful to researchers who wish to build on it in the future. However, I know I’ve suggested a lot of crazy experiments here. I would not expect all of this necessarily to be done and I leave it up to the discretion of the researchers which are most important. I am also sure the authors could come up with better ablations than these as well. But my sticking point is the first ablation — initializing with non-contextualized embeddings — which I think is critical. And I think it behooves the authors to address some of the lingering questions (including more written below), even if not all of them.

### Recommendation

Unfortunately, reject. The technique is simple and the results seem good, but the paper does not provide empirically-justified insight on why TNF works. I think ablations and investigation into the “why” aspect is the most important part of this kind of model engineering research.

### More comments & questions

I am left with some more questions about how TNF works:

* How does the quality of the representations of rare words specifically compare in your approach? Does it improve the representations of common words and contextualization at the expense of rare words? While it may be tricky to try to directly assess embedding or contextualization quality, breaking down the MLM perplexities by word frequency (or presence of rare words in the context) after removing the note dictionary might be informative. I admit this might also be tricky because I imagine the model would have to be fine-tuned without the notes for a bit before doing such an experiment. But any insight into this issue would be appreciated.
* If this method indeed works by more narrowly refocusing the training signal on the masked token than the context tokens, then would you be able to further increase the learning efficiency by oversampling rare words when determining the masks in training? I am not aware of anyone showing such a thing to work, though I might have missed it. Just a thought.

While the pretraining corpus is huge, 100 occurrences still seems like a pretty high threshold for rare words given the justification provided in the paper. Questions:
* What do the even rarer words look like? Are they just a source of noise? e.g., because they are components of names or don’t have clear and consistent semantic content?
* What proportion of contexts contain words appearing less than 100 times? It seems that the 20% figure in the paper is meant to apply to your definition of rare words, which appear between 100 and 500 times.
* What is the word vocabulary size? i.e., how many words appear more than 500 times, and less than 100?
* Did you do any preliminary experiments with other thresholds? Would you expect this to work with more common words as well? Why or why not? (This may also relate to the “too easy” vs “too hard” issue.)

On pre-training efficiency results: I think Figs 3a and 3b need to be explicitly qualified a little better. AFAICT, having lower loss here doesn’t necessarily mean the model (modulo the note dictionary) is learning better, because it sees the notes in the input. So we’re looking at the loss in a different setting than we intend to fine-tune in. It’s still interesting to see, but I think it's best to include an explicit caveat.

What about training the models for more steps? Will the trend hold and performance improve overall, or will the gains eventually level off as the representations of rare words get better? Especially for pretrained models, since they are used as the starting point for many models, it is often worthwhile to train them longer (as in the RoBERTa paper), so it’s important to understand the usefulness of this method in that regime.

### Typos etc.:

* P.3: neglectable -> negligible
* P.3: Representation -> Representations (in BERT acronym)
* P.6 Sec. 4.1: after “MNLI” there is a space missing after the period.
* P.6: “FULL-SENTENCES” would look better & be consistent with Liu et al if it were in small caps.
* Please cite the individual dataset creators for the datasets in the GLUE benchmark.

---

Update: upped score from 4 to 5; see comment thread.

Update again: score further updated from 5 to 6 with GloVe context ablations and perplexity results on sentences with rare words.

---

> ### Author Response · Authors · 2020-11-17
> **Response to AnonReviewer4**
>
>
> We notice that Reviewer 4's concern about why TNF works firstly lies in why rare word embeddings are poorly trained in BERT and if pre-pretraining word embedding methods can solve this problem. We will start from there to explain our analysis about why TNF works and present our empirical evidence.
>
> ---
> 1. Regarding the comments "the gradient flow in BERT can already lead to good rare word embeddings. " (Second paragraph in "Weakness")
>
> We respectfully disagree with the reviewer's claim that "the gradient flow in BERT can already lead to good rare word embeddings. " The learning of rare word embeddings has been a challenging problem since the first day of deep learning for NLP. Even when the input embedding and the output embedding (you mentioned softmax layer) are tied, the embedding of rare words are still very poorly optimized, see [1]. There are even recent works suggesting that rare word embeddings act as noise[2]. We have also listed several related and latest references in the related work section.
>
> ---
> 2. Regarding why TNF works.
>
>
> We think TNF works because it can benefit the BERT model as a whole by providing a more effective data utilization. The reasons behind are as follows.
>
> According to many previous works (see the related work section), rare words' embeddings are usually poorly optimized. There are even recent works suggesting that rare word embeddings act as noise[2]. Training a model from noisy inputs is less effective in general (see [3](Figure 1a) for more observations about how noisy inputs hurt the optimization and generalization of a machine learning model). For language pre-training tasks specifically, the input sequences with noisy rare word embeddings contain less semantically meaningful information for the Transformer to learn, making the whole model training ineffective.
>
> Given the above facts, we aim to reduce the noise from rare word embeddings in a sentence, to improve the pre-training of the whole model. Specifically, we achieve this by providing more precise semantics of rare words in a sentence using surrounding contexts from other sentences (contextualized information saved in notes). For the original method, Transformer receives thousands of noisy embeddings like 'Covid-19' containing little semantic meanings. Suppose we equip 'Covid-19' with a note that contains previous surrounding contextual information such as 'pandemic' and 'global crisis' (which are popular words). One could imagine that the sentence 'Covid-19 (+pandemic + global crisis) causes thousand of lives' could have more precise semantics, which makes the training of the Transformer model more effective.
>
> ---
> 3. Empirical evidences for supporting 2.
>
> To support such a claim, we conduct a new experiment as below:
>
> we calculate MLM loss on sentences in the validation set that do not have rare words for both BERT and BERT-TNF at different checkpoints. The total number of sampled sentences that satisfy the condition is roughly 20k.  As those testing sentences don't contain rare words, the note dictionary will not be called, and we can purely compare the performance between the two Transformer models in BERT and BERT-TNF. From the table below, we can see that BERT-TNF's loss is consistently lower than BERT at all checkpoints, which suggests the entire Transformer model was improved using our method.
>
> #Iter 20k 50k 100k 200k 400k 600k 800k 1000k
>
> BERT 2.709 2.180 1.947 1.788 1.667 1.602 1.556 1.522
>
> TNF 2.692 2.145 1.902 1.736 1.619 1.558 1.513 1.479
>
> In the submission, we also showed that TNF without notes works well on finetuning downstream tasks, indicating that TNF can provide a better pre-trained model itself (without notes saved in downstream tasks). Moreover, even for sub-tasks with almost no rare words occurring in the training set (0.47% rare word coverage in CoLA), BERT-TNF can still outperform BERT on it.
>
> All empirical results above support our motivation and indicate that the model as a whole (including the Transformer layers and token embeddings) is better pre-trained with TNF.
>
> ---
> 4. Regarding the comment "the notes of rare words could 'explain away' rare word embeddings." (Third paragraph in "Weakness")
>
> We respectfully disagree with Reviewer 4 that the notes of rare words could 'explain away' rare word embeddings. Given that rare-word embeddings are already poorly-trained and act as noise, their quality has little margins of being hurt. That being said, adding notes for rare words does not 'explain away' from rare word but 'explains' the rare word better. It is because as illustrated above, after adding notes, the input sentence with this rare word contains more accurate semantics about the rare word for the model to learn.

---

> > ### Author Response · Authors · 2020-11-17
> > **(Following the previous comment)**
> >
> > ---
> > 5. Regarding whether easy/hard tasks relate to training efficiency. (Forth paragraph in "Weakness")
> >
> > We respectfully disagree with the reviewer that a task being 'too easy' or 'too hard' is a proper metric to evaluate if it would be effective for language pre-training.
> > A task will be more effective for the Transformer to learn if it is more semantically meaningful, so that the model can be guided to learn from more informative signals in the training corpora.
> >
> > For example, ELECTRA[4] does that by making the model to discriminate more semantically ambiguous replaced tokens. PMI does that by providing a more semantically meaningful masking instead of forcing the model to predict meaningless tokens such as 'ness' or 'ing' after BPE. TNF's task is also more semantically meaningful compared with BERT. Because TNF provides a more semantically reasonable data utilization by using notes, to reduce the noise of the input sentences caused by rare words. That is the reason why TNF can further improve ELECTRA when applied on top of it. We also believe TNF can be combined with PMI together for further acceleration.
> >
> > We would also like to note that, if a harder task is always better for language pre-training while an easier task is always more harmful, then it is impossible to explain why Electra is significantly better than BERT. Because ELECTRA actually changes BERT's multi-classification task of predicting the correct word into a binary classification problem of discriminating replaced tokens. A binary classification task should be much easier than a multi-classification task with 30k+ classes.
> >
> > ---
> > 6. The ablation studies that R4 suggested to show why TNF works.(after 4th paragraph in "Weakness")
> >
> > First of all, we would like to note that the reason why TNF works is explained in bullet 2 with additional empirical evidences. While we do notice that we have run several of ablations that you suggested before the submission. Therefore we present the results here.
> >
> > Ablation study 1. Initializing BERT word embeddings with pre-pretrained w2v or GloVE embeddings.
> >
> > We conducted an experiment to initialize the embedding using word2vec, and train BERT further with respect to MLM loss. We found this model performs almost identical to BERT. Note that w2v or GloVE cannot improve the quality of rare word embeddings (actually they have this problem since day one, see related work in the submission), let alone improving language pre-training methods on top of it.
> >
> > Ablation study 2. Directly update the rare word’s embeddings with a version of Eq. 5.
> >
> > This actually means that you are setting the $\lambda$ to be 1 in Eq. 6 and still keeps the positional embedding. Because by doing so you basically ditched the rare word embeddings which are optimized normally with the MLM loss. We haven't tried setting $\lambda$ to be 1, but we have tried setting $\lambda$ as 0.9 in the submission (See Table 4 in the appendix, R9). The result shows that with $\lambda$ being 0.9 TNF performs similar (only 0.1 minus) with the optimal TNF.
> >
> > Ablation study 3. Update the note embeddings via backprop instead of Eq. 5.
> >
> > If we understand correctly, you suggest keeping a note embedding emb1 for a rare word together with its word embedding emb2. The two embeddings are always used together (emb1+emb2) and updated by gradient descent. This is equivalent to setting an emb' = emb1+emb2 and directly update emb', which is further equivalent to the original BERT. Therefore the result of this ablation is predictably identical with that of BERT.
> >
> > ---
> > 7. Regarding the quality comparison between rare/common word embeddings in TNF (The first bullet point under 'More comments & questions').
> >
> > We would like to note that language pre-training is not just a game of word representations. The amount of Transformer parameters is significantly more massive than that of word embeddings, and the Transformer model significantly improves isolated word embedding by contextual embeddings. As illustrated in 2. and 3., TNF does not aim to strengthen word embeddings with a particular range of frequency, but to improve the model as a whole.
> >
> > ---
> > 8. Regarding oversampling rare words (The second bullet point under 'More comments & questions')
> >
> > ELECTRA[4] has tried that before (dynamically masking more rare words) and it doesn't lead to any performance improvements.

---

> > > ### Author Response · Authors · 2020-11-17
> > > **(Following the previous comment)**
> > >
> > > ---
> > > 9. Regarding questions about the occurrence range [100-500].
> > >
> > > Thanks for this question. We have tuned this frequency range to build the note dictionary before the submission. We apologize for not having put the results into the submission. Specifically, we have tried setting the range to be 50-100 and 100-1000.
> > >
> > > Range 50-100 performs almost identical to the BERT baseline. The reason is that rare words with numbers of occurrences between 50-100 can only cover 0.3% of the whole training corpora. With such low coverage, very limited training sentences would be enhanced by rare word notes. As a comparison, words whose occurrences between 100-500 cover 2% of the training corpora, which is shown empirically that could impact the pre-training. We cannot further include lower-frequency (e.g., < 50) words in the dictionary as the number of such words would exponentially increase due to the long-tail problem of language corpora, occupying a massive amount of GPU memory.
> > >
> > > Range 100-1000 performs similar to our main results from 100-500, which suggests the rare word range in TNF is robust to some extent.
> > >
> > > The total size of word vocabulary is 9435484, while the number of words with occurrences under 100 is 9163284. Given the massive amount of words under 100, we can't check all of them. While we do agree with the reviewer that there are indeed some components of names or don’t have clear semantic meanings and they could act as noises.
> > >
> > > ---
> > > 10. Regarding Figure 3.
> > >
> > > Thanks for this question. We agree that the lower MLM loss during pre-training we report in Figure(3) can be brought by both the extra memory and the better Transformer parameters. While according to the new additional experiment above, we can see that TNF's MLM validation loss on sentences with no rare words is also consistently lower than that of BERT, which indicates that the BERT model itself is also better-trained.
> > >
> > > ---
> > > We appreciate Reviewer 4's active comments despite the disagreements we illustrate earlier. Please do let us know if we understand your comments correctly and if we have addressed your concerns.
> > >
> > >
> > >
> > > ---
> > > References.
> > >
> > > [1] Gao, Jun, et al. "Representation degeneration problem in training natural language generation models.", (ICLR 2019).
> > >
> > > [2] Li, Yangming, et al. "Handling Rare Entities for Neural Sequence Labeling.", (ACL 2020).
> > >
> > > [3] Zhang, Chiyuan, et al. "Understanding deep learning requires rethinking generalization."(ICLR 2017).
> > >
> > > [4] Clark, Kevin, et al. "Electra: Pre-training text encoders as discriminators rather than generators.", (ICLR 2020).

---

> > > > ### Comment · AnonReviewer4 · 2020-11-19
> > > > **Extra results are helpful, but the gap still remains to be explained**
> > > >
> > > > Thank you for your active and engaged reply as well. I will address the points in order.
> > > >
> > > > 1. On whether "the gradient flow in BERT can already lead to good rare word embeddings."
> > > >
> > > > I did not say this, and I am not sure where you got this quote from. My point was that there is an apparent tradeoff between the learning signal on the rare word embeddings and the learning signal on the rest of the network. This does not mean that the rare word embeddings will be "good," though intuitively we would expect them to be better when there is more learning signal on them.
> > > >
> > > > Actually, your reference [1] on this point proposes something very interesting: that the degeneration of the embeddings of rare words results from their use in the *output* layer. That suggests that the real problem here might be weight tying between inputs and outputs, which indicates that the key component of the TNF might indeed be that it decouples the input embeddings from the learning signal at the output layer. This is exactly what I suggested investigating in the second ancillary ablation, to see how much of the benefit comes from untying the weights versus the specific method of updating the note dictionary. Also note that while ELECTRA does binary classification, its embeddings are tied to its generator model during pretraining, which does use the MLM objective, so still may have this problem.
> > > >
> > > > 2, 3. On rare word embeddings as noise, and evidence that TNF mitigates this
> > > >
> > > > Upon random initialization, all embeddings start as "noise." Then they are learned. Or, in the case of rare words, perhaps they are not learned, or not learned well. It certainly seems that TNF helps mitigate this problem. I believe this from the experiments in the paper and in your point 3 (thanks for that). But many questions remain about *how* and *why* TNF mitigates it, which is the principal complaint in my review.
> > > >
> > > > Your explanation with the COVID-19 example is insufficient to explain why TNF would work but simpler methods wouldn't: particularly, initializing with GloVe (or similar), or just updating the rare word embeddings directly instead of having a separate note dictionary. Additively incorporating information from nearby tokens in previously observed context windows is *exactly* how the update rules in methods like GloVe and word2vec work. So going just by your explanation, there is no reason to think the representations in TNF's note dictionary will be any higher quality than GloVe or word2vec representations. And yet, TNF works better, as you say in your point 6.1. That means there is something about TNF beyond the reasons you give that is responsible for why it works. The point of the ablations I suggest is to get at what the real difference is that matters. In fact, that brings another ablation idea to mind, which is to use fixed GloVe embeddings as the note embeddings. (I do expect this to perform worse than TNF, but it would be very informative to see just how much worse).
> > > >
> > > > 4. "explaining away"
> > > >
> > > > My point here was about the learning signal on the rare word embedding (not including the note embedding) which is retained in fine-tuning. If this has less learning signal (which it well, due to the "explaining away" effect of including the note embedding), we would expect it to be even worse quality at the end of training, especially if the low quality is related to its use in the output layer. Maybe the effect isn't large because the embedding was already bad, so the tradeoff works out — but there is still a tradeoff here. I don't see the point of denying that.

---

> > > > > ### Comment · AnonReviewer4 · 2020-11-19
> > > > > **(cont.)**
> > > > >
> > > > > 5. Quality of learned representations when the task is easy / hard / semantically meaningful.
> > > > >
> > > > > I did not suggest that harder tasks are always better for pretraining and easier are always worse. I explicitly said that it seems that it's possible for a task to be too hard (i.e., not learnable and thus indistinguishable from noise to the model) *or* too easy (i.e., leading the model to learn simple shortcuts instead of generalizable semantics). The "too easy" issue is the same as the "explaining away" issue — if a simple shortcut suffices, the model is less incentivized to learn anything more sophisticated. So in this case the thing to explain is why reducing signal on the rare word embeddings doesn't harm them enough to hurt performance in fine-tuning.
> > > > >
> > > > > I don't think your argument about the task being "semantically meaningful" is relevant to TNF. You could just as easily view the case of rare words without TNF as a "semantically meaningful" testbed through which to learn embeddings for rare words; in this case, TNF reduces the "semantically meaningful" aspect by the explaining away effect. The point here is that learning the relationship between a rare word and its context was hard enough not to be efficiently learnable by the model, so you added a note dictionary and update rule to make it easier. Indeed, this may mean that *less* is learned about relating rare words to their context than in the non-TNF case, but making that subproblem easier facilitates learning in the *rest* of the model, as you argue, and this is ultimately more important for downstream performance.
> > > > >
> > > > > 6. Ablations
> > > > >
> > > > > Thanks for sharing your results on the first two. However, be careful about the claim that word2vec or GloVe cannot learn good representations for rare words. If that is the case, then it would naturally follow that TNF couldn't either, since the update rules are so similar. You would need to explain what the difference is that matters.
> > > > >
> > > > > On point 6.3: No, the embeddings are not always used together. If I understand correctly, TNF only brings in the note embeddings in the input layer, and not the output. This ablation is very close to (and can be substituted with) simply untying the word embeddings of the rare words in the input and output layers. As I said above, your reference [1] seems to indicate this might be a good idea, at least for BERT.
> > > > >
> > > > > 7. I wasn't saying that TNF has to improve all of the embeddings. Rather, it seems to me it would make embeddings of rare words worse. It would be interesting to know if this is the case — that's all. Your results you provided in point 3 seem to answer the question for common words, which is very informative on that end. Thanks.
> > > > >
> > > > > 8. Ah, thanks for pointing that out. I was not aware of this experiment (just found it in the ELECTRA paper's appendix). I was curious about whether TNF would make such a thing work better but I guess it seems unlikely and it isn't important.
> > > > >
> > > > > 9. Word vocabulary and frequency cutoffs — I see. Makes sense; the tail is just way too long because we're working with words and not wordpieces. Thanks.
> > > > >
> > > > > 10. Sounds good. I think your results from point 3 would be very nice to include in the paper.
> > > > >
> > > > > ---
> > > > >
> > > > > Anyway, the results in point 3 and for word2vec/GloVe initialization are helpful. I think these improve the paper's case. However, particularly because word2vec/GloVe didn't perform any better than BERT, the explanation in the paper for why/how TNF works is quite inadequate. More experiments are needed. For this reason I am still recommending rejection, though I will up my score to 5.

---

> > > > > ### Author Response · Authors · 2020-11-21
> > > > > **Second round of response to AnonReviewer4**
> > > > >
> > > > > Thanks for the quick response and clarification of the comments! We find that you have two remaining concerns in general: why using GloVe (w2v) doesn't work while TNF works, and whether the rare word embedding is improved or not. We provide the answers and new empirical analysis below. Please correct us if we misunderstand your comments.
> > > > >
> > > > >
> > > > > 1. Why using BERT contextual outputs around rare words as notes would be better than simply using pre-trained GloVE or w2v rare word embeddings as notes and how much better?
> > > > >
> > > > > If we understand it correctly, your concern behind this question is that ”Additively incorporating information from nearby tokens in previously observed context windows (the way to update notes in TNF) is exactly how the update rules in methods like GloVe and word2vec work“.
> > > > >
> > > > > First of all, the surrounding information of rare words that TNF uses is much better than Glove or w2v rare word embeddings. TNF leverages the surrounding information using the BERT's output representations, which are significantly superior to conventional word embeddings due to BERT's powerful semantic representation capacity. (If BERT contextual outputs are of the same quality or just slightly better than w2v word embeddings, then people would have kept building models on top of pre-trained word embeddings instead of switching to BERT.) Therefore compared to w2v, using BERT's contextual outputs as notes can provide more accurate signals and help the model train better.
> > > > >
> > > > >
> > > > > To verify this and answer the 'how much better' question, we have run the ablation experiment that you suggested, using fixed GloVe rare word embeddings as the notes. We cannot finish the whole pre-training before the rebuttal deadline due to its long-running time. But both the current train loss and validation loss curve (before around 200k iterations) of this ablation are slightly lower than BERT while still significantly higher than BERT-TNF. It supports our argument above.
> > > > >
> > > > >
> > > > >
> > > > > 2. Are the rare word embeddings trained with TNF worse or better than those of BERT?
> > > > >
> > > > > We apologize for misunderstanding your original comments related to this. In order to answer this question, we calculate the MLM validation loss on validation sentences containing rare words. In particular, we evaluate the BERT-TNF models with/without notes, and compare it with the BERT baseline at different checkpoint. Here are the results.
> > > > >
> > > > > #Iter 20k 50k 100k 200k 400k 600k 800k 1000k
> > > > >
> > > > > TNF-with notes 3.896 3.100 2.745 2.504 2.333 2.246 2.179 2.131
> > > > >
> > > > > TNF-without notes 3.929 3.170 2.850 2.633 2.467 2.381 2.313 2.265
> > > > >
> > > > > BERT 3.921 3.153 2.815 2.583 2.408 2.313 2.246 2.197
> > > > >
> > > > >
> > > > > From the results, we can see that on the rare-word sentence, the performance order is BERT-TNF with note > BERT > BERT-TNF without note. We can see that first, BERT-TNF with note achieves the best performance. Interestingly, BERT is better than BERT-TNF without note. From the previous experiment on popular sentences, we conclude that BERT parameter is better trained using TNF. The new observation may lead to a conclusion that: the rare word embedding trained using BERT-TNF is worse than that trained using BERT. We think it can act as strong evidence showing that TNF indeed splits out some rare word information into the notes as you hypothesized.
> > > > >
> > > > >
> > > > >
> > > > > We would like to thank Reviewer 4 again for the efforts on reviewing this submission and conducting very active discussions. We will include our discussion in the next version and try our best to update the pdf as soon as possible.

---

> > > > > > ### Comment · AnonReviewer4 · 2020-11-22
> > > > > > **The new results sound great**
> > > > > >
> > > > > > Thanks for your response.
> > > > > >
> > > > > > 1. On the quality of the context vectors — Yes! I agree that BERT's contextualized vectors intuitively should be quite a bit more informative than non-contextualized GloVe vectors for these purposes. The experiment seems to confirm this intuition. Thank you!
> > > > > >
> > > > > > 2. On the quality of the rare word embeddings — Yes again! This is also what I suspected and validates the intuition that you are essentially giving up some of the learning signal on the rare word embeddings in order to improve the signal on the rest of the model. I felt that this is what the paper was basically getting at but fell short of saying outright. These experiments are starting to give a more explicit accounting of what is going on in TNF.
> > > > > >
> > > > > > I think these experiments put the paper over the publication threshold. I'm upping my score again.
> > > > > >
> > > > > > Given the results that you've shared, I'm still pretty curious about one of the remaining ablations: optimizing the notes via backprop. I assume it should be very simple to implement, so I hope you are able to run it (assuming you have the resources). If you can, I think it would further strengthen the case for this paper. You've shown that the TNF update rule with contextualized vectors works better than GloVe, but we still haven't seen whether it works better than backprop (at least, without the potentially destructive influence of the output layer). There are two possible outcomes:
> > > > > >
> > > > > > 1. It works as well as TNF. In this case it might be preferable since it's a bit simpler, and it would also point to some potential problems with BERT: 1) problems with the loss on the output layer may be holding the model back, or 2) the wordpiece tokenization method makes it hard to learn good representations of composite/rare words. Disentangling those two contributors would be future work (though I think there is some work which has hinted towards the second). There may also be other interpretations I haven't thought of.
> > > > > >
> > > > > > 2. It *doesn't* work as well as TNF. Then the reason that comes to mind is that backprop is not a great update rule when gradients on a parameter are extremely sparse, either because there are too few updates to get it into a good part of the space at all, or because the updates are made so sparsely during training that the rare word's representation lags behind the latent space learned by the rest of the model. These questions could be further tested by 1) some kind of clever initialization with GloVe (though this might be a bit weird for the sub-word wordpieces), or 2) using low-rank or computed embeddings from a method like DeFINE (https://arxiv.org/abs/1911.12385) which doesn't have the problem of sparse updates. If none of those methods work either, then this would mean TNF provides strong evidence in favor of more generally combining backprop with discrete update rules like TNF's equation (5) when dealing with parameter sets that have extremely sparse gradients. This would suggest some very intriguing directions for future work.
> > > > > >
> > > > > > Anyway those are just some thoughts I wanted to share on interpreting the results — besides the backprop ablation the rest of these experiments seem out of scope to me.
> > > > > >
> > > > > > I also think Reviewer 1 makes a good point about model capacity. Recent work suggests that in the very-large-model regime, more parameters may help a model learn faster (https://arxiv.org/pdf/2002.11794.pdf). So TNF may be working partly by just increasing the effective capacity of the model. A fairer comparison *may* be to compare BERT+TNF to a version of BERT enlarged to have a similar number of parameters, and I think such an experiment might further strengthen the paper — particularly if the argument is about speeding up training under a memory budget. (TNF has the advantage of not needing the extra memory at fine-tuning time, but perhaps the disadvantage of worse representations of rare words, which could hurt performance depending on domain, following along with Reviewer 1's comments.)  However, I think the experiment you shared here with GloVe is mostly convincing that a lot of the value is indeed provided by the TNF update rule anyway, so this doesn't seem like a deal-breaking observation to me.

---

> > > > > > > ### Author Response · Authors · 2020-11-24
> > > > > > > **Thanks for your advice and appreciating our reply**
> > > > > > >
> > > > > > > Your comments have indeed made us curious about what can actually happen if we optimize the notes via backprop. We will run this ablation and take a look at the gradients.

---

### Official Review · AnonReviewer1 · 2020-10-29
**The method is simple and seems to be very effective in a certain situation, but we do not know what that situation is and why**

**Rating:** 6
**Confidence:** 3

**Review:**

The paper proposes an external memory architecture. When encountering the rare words (with a frequency between 100-500), the method will store the average contextualized word embedding of nearby words into a dictionary. Next time it encounters the same rare word, it will retrieve the average embedding and input it into BERT encoder. The experiment results show that given the same number of training steps, adding the external memory improves the MLM loss and significantly improves the results on RTE (Recognizing Textual Entailment) dataset, which leads to a slightly better GLUE score. The experiment also shows that keeping the external memory during the fine-tuning stage slightly degrades the performance.

Pros:
1. The method is simple and easy to understand
2. The experimental results on GLUE are quite surprising. It shows that we should take note when training BERT but throw away the note dictionary when fine-tuning the model.

Cons:
1. Missing an important citation [1]
2. The paper does not well explain the surprising results on GLUE. This is a crucial weakness. The comparison of the MLM loss is not very fair because the proposed method has a large external memory. The benefit of the proposed method relies on the improvement of the average GLUE score. However, Table 2 shows the most of the improvement of GLUE actually comes from the improvement of a single dataset, RTE. Without understanding why it improves RTE, the readers do not know when they want to adopt the proposed method for their downstream applications.

Clarity:
The text is fluent, but the main story is not well supported by the experiment results. The story is that using an external dictionary could accelerate the training, but the main experiment finding actually says that using an external dictionary can very significantly improve the results on RTE dataset while performing similarly on other datasets in GLUE.

Originality:
There has been some effort of using an external dictionary to help the training of BERT [1], but I am not aware of existing papers that apply the dictionary to only the rare words. I also do not know any other work that shows the external dictionary could improve the GLUE scores.

Significance of this work:
If the authors could well explain the experimental results on GLUE and justify the explanation using some analysis, this might lead to more important findings.


Figure 3c seems to contradict with Table 1 and 2 because in Table 1 and 2, the GLUE score of BERT (ours) is 83.1 but all the points in the BERT curve in Figure 3c is below 83.

Usually, when a study tries to sell its method as a way to accelerate the training, it means the method reaches some performance faster but the method will converge the same performance eventually. However, Figure 3 does not show that they will converge the same value, so selling the method as a way to accelerate the training is weird. Furthermore, I think the lower MLM loss is due to the extra parameters in the note dictionary rather than the note dictionary accelerates the training.

It is not surprising that taking notes for rare words could achieve lower loss/perplexity because the note dictionary gives the extra memory capacity [1]. It is also not surprising that it can achieve better performance on GLEU if using the note dictionary during the fine-tuning stage due to the extra parameters. The really interesting results are that the authors report that the model could very significantly improve the RTE task and mildly improve CoLA without using the note during the fine-tuning stage.

Intuitively, the proposed model stores lots of knowledge about the rare words into the note dictionary. Does the fact that the note is not needed in the fine-tuning stage imply that the knowledge about rare words is actually not needed? Does it mean the RTE or CoLA do not contain many rare words or does it mean the rare words do not affect the decision of BERT and ELECTRA in RTE or CoLA? Is the reason of improvement that we could store more interactions between popular words in the parameters of BERT itself because the information of rare words has been stored in the note (maybe you can test this by reporting the MLM loss on the sentences without any rare words)? If that is the case, why do we only stably improve RTE and CoLA? If the authors can show the above hypothesis is true, I think this is a significant contribution because that means this paper provides a way to control what LM should learn when there is a mismatch between MLM training corpus and downstream applications (e.g., MLM training corpus contains many rare words but we should ignore the rare words in the downstream applications).

This paper lacks a good explanation of the above weird result (in my opinion, the most valuable finding in this paper) and lacks the analysis that supports the explanation. The main paper says that taking notes improves the tasks with the small datasets the most. The STS-b (7k) and MRPC (3.7k) have smaller training datasets than CoLA (8.5k). Why are the results of STS-b and MRPC cannot be stably improved? If the authors really want to explain the performance improvement using the training dataset size, the authors can just randomly sample several small subsets of training data from each dataset and show that the GLUE score improves a lot in that setting. In the appendix A.4, the authors hypothesize that the small proportion of rare words in each dataset of GLUE (from 0.47% to 2.31%) might be the reason that we can ignore the note dictionary during the fine-tuning stage. This also did not explain why most of the improvement of the GLUE score comes from RTE. Moreover, if the rare words are not important in the testing datasets, why do we want to take notes in the first place?

I will vote for acceptance if the authors could answer these critical questions I raise above strongly.


Minor:
1. Although the chance is not high, I think it is possible that parts of MLM improvement could be achieved by simply sampling the sentences containing the rare words more (This is a minor concern. If you do not have time to finish the experiments for this baseline, you can choose not to do it or compare the results after training fewer steps).
2. I guess the dictionary overhead is small but it should be measured and reported because you say the method accelerates the training.


[1] Lample, Guillaume, et al. "Large memory layers with product keys." Advances in Neural Information Processing Systems. 2019.

---

> ### Author Response · Authors · 2020-11-17
> **Response to AnonReviewer1**
>
>
> We would like to thank Reviewer 1 for the constructive comments and careful reading. From your comments, we realize that we didn't describe our motivation clearly. It leads to some difficulties in reading and further leads to some critical misunderstandings of our work. We first describe why our method works and present more empirical evidence, which we think can largely help better understand our paper. Then we answer each question separately.
>
> ## The motivation of TNF
>
> First of all, we would like to emphasize that poor rare word embeddings will hurt the training of all model parameters (such as the Transformer layers). The reasons behind are as follows. According to many previous works (see the related work section), rare words' embeddings are usually poorly optimized. There are even recent works suggesting that rare word embeddings act as noise[1]. Training a model from noisy inputs is less effective in general (see [2](Figure 1a) for more observations about how noisy inputs hurt the optimization and generalization of a machine learning model). For language pre-training tasks specifically, the input sequences with noisy rare word embeddings contain less semantically meaningful information for the Transformer to learn, making the whole model training ineffective.
>
> Given the above facts, we aim to reduce the noise from rare word embeddings in a sentence, to improve the pre-training of the whole model. Specifically, we achieve this by providing more precise semantics of rare words in a sentence using surrounding contexts from other sentences (contextualized information saved in notes). For the original method, Transformer receives thousands of noisy embeddings like 'Covid-19' containing little semantic meanings. Suppose we equip 'Covid-19' with a note that contains previous surrounding contextual information such as 'pandemic' and 'global crisis' (which are popular words). One could imagine that the sentences  'Covid-19 (+pandemic + global crisis) causes thousand of lives' could have more precise semantics, which makes the training of the Transformer model more effective.
>
> We notice that Reviewer 1 has also reached a similar understanding (`` we could store more interactions between popular words in the parameters of BERT itself'') and ask for more supporting empirical evidence. We follow your advice to conduct the experiment below.
>
> We calculate MLM loss on sentences in the validation set that do not have rare words for both BERT and BERT-TNF at different checkpoints. The total number of sampled sentences that satisfy the condition is roughly 20k. As those testing sentences don't contain rare words, the note dictionary will not be called, and we can purely compare the performance between the two Transformer models in BERT and BERT-TNF. From the table below, we can see that BERT-TNF's loss is consistently lower than BERT at all checkpoints, which suggests the entire Transformer model was improved using our method.
>
> #Iter        20k       50k       100k         200k        400k       600k     800k     1000k
>
> BERT       2.709    2.180     1.947       1.788       1.667      1.602    1.556     1.522
>
>  TNF         2.692    2.145     1.902       1.736       1.619     1.558     1.513     1.479
>
> In the submission, we also showed that TNF without notes works well on finetuning downstream tasks. Moreover, even for sub-tasks with almost no rare words occurring in the training set (0.47% rare word coverage in CoLA), BERT-TNF can still outperform BERT on it.
>
> All empirical results above support our motivation and indicate that the entire model is better pre-trained with TNF.
>
> ## Response to other questions
>
> * Regarding the imbalanced performance improvements of TNF
>
>
> Thanks for the careful checking. We think the reason for this imbalanced improvement gain is that sub-tasks in GLUE have different margins for improvements in general. For example, BERT-Large is three times larger than BERT-Base. Its improvements over BERT-Base on RTE and CoLA is more than 3 points. While for the rest of tasks like MRPC and STS-B, their performance gaps are relatively small, e.g., 0.4 and 0.7. This indicates that some tasks, like RTE and CoLA, have a larger improvement space when the model is more powerful. While for other tasks, the improvement space may be limited. Similar trends can also be found in other language pre-training methods such as SpanBERT[4] and ELECTRA[5].
>
> We understand that imbalanced performance improvements look weird when readers have concerns about why TNF works. While given that it is a common trend of a lot of other language pre-training methods, we think this phenomenon is orthogonal to TNF.

---

> > ### Author Response · Authors · 2020-11-17
> > **Following the previous  Comment**
> >
> > * Regarding selling TNF as an acceleration method
> >
> > We agree that ideally, we need to check the 'converged' point. However, language training methods such as BERT are severely under-trained[6]: as BERT data is huge, we usually can not observe the valid loss ``converge'' even for one-week training. Given such a situation, we present the performance (pre-training and fine-tuning) given the same computational budget (iterations) as evidence.
> >
> > We agree that the lower MLM loss during pre-training we report can be brought by both the extra memory and the better Transformer parameters. According to the additional experiment above, we can see that TNF's MLM validation loss on sentences with no rare words is also consistently lower than that of BERT, which indicates that the BERT model itself is also better-trained.
> >
> > * Regarding the inconsistency between Figure 3c and the tables
> >
> > Thanks for the careful checking and we apologize for the inconsistency. Results in Table 1 and 2 are the correct results we wanted to present. When we run the last batch of hyper-parameter tuning experiments to update the main results, we forgot to update the figure in the last minute. We will fix it in the upcoming version of the submission.
> >
> > * Regarding the missing citation
> >
> > We will add the citation in the upcoming version of the submission.
> >
> > ---
> > Reference.
> >
> > [1] Li, Yangming, et al. "Handling Rare Entities for Neural Sequence Labeling.", (ACL 2020).
> >
> > [2] Zhang, Chiyuan, et al. "Understanding deep learning requires rethinking generalization."(ICLR 2017).
> >
> > [3] Devlin, Jacob, et al. "Bert: Pre-training of deep bidirectional transformers for language understanding.", (ACL 2019).
> >
> > [4] Joshi, Mandar, et al. "Spanbert: Improving pre-training by representing and predicting spans.", (ACL 2020).
> >
> > [5] Clark, Kevin, et al. "Electra: Pre-training text encoders as discriminators rather than generators.", (ICLR 2020).
> >
> > [6] Liu, Yinhan, et al. "Roberta: A robustly optimized bert pretraining approach." arXiv preprint arXiv:1907.11692 (2019).
> >
> > ------
> > We would like to express our appreciation again to Reviewer 1's critical questions, which greatly help us polish our work. We will include the above discussions in the rebuttal version of the submission. We hope our response can address your concerns, and please let us know if you have any further questions.

---

> > ### Comment · AnonReviewer1 · 2020-11-18
> > **The new results look good**
> >
> > First, although I still think it would be better to know why TNF improves RTE more than CoLA, I think your explanation is reasonable.
> > Second, I think the new results you provide are significant. Like I said in the original comment, I think the results show that TNF could become a way of controlling what LM should learn. For example, if you want to train BERT on Wikipedia but your downstream tasks do not contain any sentences in the biomedical domain, you can alleviate the mismatch by taking notes for biomedical terminology and remove the note during testing time. Just like you find that rare words are indeed rare in downstream tasks, so taking notes for rare words during training allows BERT to spend more memory on the interactions between other words. I will change my vote to 6 for this contribution.
> >
> > As for the explanation about why TNF works better, I still think the story you try to sell lacks support. We should discuss the cases of containing rare words and the case of not containing rare words separately.
> > For the case not containing rare words, it is possible that the improvement comes from better modeling the interactions between words as you said. It is also possible that the improvement comes from the fact that BERT could spend more memory on the sentences without rare words with the help of the note dictionary like I said.
> > For the case containing rare words, it is possible that the loss improvement comes from better modeling the interactions between words or avoiding the noise as you said. It is also possible that the improvement comes from the extra memory in the dictionary like I said.
> > We need more analyses to tell us which explanation is more true (or both of them are equally true). Some possible analyses include you can test the MLM loss without using the note dictionary (as you did in the GLUE benchmark), and you can just sample the sentences containing the rare words more frequently in the baseline (i.e., not using TNF) to improve the quality of the rare word embeddings. These two suggestions are just some random ideas. Feel free to ignore them if you have a better idea. I understand that it is difficult to design the experiments to investigate which explanation is better. If the authors can use more analyses to better support the story of the paper, I will further boost my score to 7.

---

> > > ### Author Response · Authors · 2020-11-21
> > > **Second round of response to AnonReviewer1**
> > >
> > >
> > > Thanks for the quick and enlightening response! Regarding the new comments, we provide more results and analysis.
> > >
> > > First, we notice you have concerns about where the improvement over sentences without rare words comes from. We are not sure what you exactly mean by 'the model spend more memory on the sentences without rare words'. If it means the model can better 'memorize' the data during training, then it is very unlikely to be the reason. This is because in the 'sentence without rare words' experiments in our rebuttal, we did the evaluation on the validation set instead of the training set. Note that the model never sees the validation sentences during training, let along memorize them. In machine learning literature, better performance on validation set usually means 'the modeling is better'. If a model merely 'memorized' more of the training data  due to the extra space while the modeling is not improved, it doesn't necessarily lead to better generalization (validation performance) (see [2] in the first round of rebuttal).
> > >
> > > Second, for cases with rare words, we have some new empirical results. We calculated the MLM validation loss of validation sentences containing rare words on BERT-TNF without using notes, and compare it with that of BERT and BERT-TNF with notes. Here are the results.
> > >
> > > #Iter 20k 50k 100k 200k 400k 600k 800k 1000k
> > >
> > > TNF-with notes 3.896 3.100 2.745 2.504 2.333 2.246 2.179 2.131
> > >
> > > TNF-without notes 3.929 3.170 2.850 2.633 2.467 2.381 2.313 2.265
> > >
> > > BERT 3.921 3.153 2.815 2.583 2.408 2.313 2.246 2.197
> > >
> > > We find that on sentences with rare words, the performance order is BERT-TNF(with notes) > BERT > BERT-TNF(without notes). We can see that first, BERT-TNF with note achieves the best performance. Interestingly, BERT is better than BERT-TNF without notes. We think it can act as a strong evidence showing that TNF indeed splits out some rare word information into the notes as you suggested.
> > >
> > > As a summary, based on the results and discussion above, we think that for TNF, whether to use rare word notes at fine-tuning should be downstream task-dependent. For normal downstream tasks such as GLUE that contain little rare words, TNF without notes works well because it helps better modeling the common word interactions as we said. While for special downstream tasks such as biomedical terminology as you mentioned, if the task's data contains too much rare words, then using notes with TNF at fine-tuning would probably be the best choice.
> > > However, we would like to note that the goal of the paper is to improve pre-training efficiency. TNF can achieve this goal because splitting rare word information into notes can help reduce the noise of the input sequence therefore make the pre-training of the Transformer better. It is irrelevant with whether or not to use notes at fine-tuning. We will include this discussion into the paper.
> > >
> > > We also really appreciate the reviewer's domain adaptation example. It has made us realize that what TNF can do is not just about rare words. It could have more potentials to address domain shift problems in NLP.
> > >
> > > We hope our explanation and new experimental results can address your concerns.

---

> > > > ### Comment · AnonReviewer1 · 2020-11-21
> > > > **Memorization clarification**
> > > >
> > > > I do not mean that Transformer will memorize the text in the training corpus word by word. What I mean is that you need a significant amount of parameters/memory in the language model in order to achieve good perplexity. For example, in the validation corpus, you might sentence like "In 2001, the New York City mayor [Mask] [Mask] handles 911 attack well". In the training corpus, some sentences might mention that the mayor is Rudy Giuliani in 2001. In order to achieve low perplexity in that validation sentence, the model needs to memorize the fact [2]. That is why the larger model will achieve lower perplexity and the model size of GPT3 is so large. [3] further discovers that the perplexity of LM depends on mostly the model size rather than the hyperparameters such as hidden state size. [1] also shows that extra memory capacity could achieve lower perplexity. All of them talk about validation less. That is why I and other reviewers said that it is not surprising that TNF could achieve lower perplexity by using extra parameters. Your new experiment results also support this hypothesis.
> > > >
> > > > My hypothesis is that most of the improvement on GLUE does not come from reducing the noise or modeling the word interaction in general as you claim. Instead, the improvement simply comes from the fact that BERT itself (without note dictionary) in TNF spends less memory on rare words, so the saved memory power could be used to memorize the facts involving popular words and interactions between popular words. Such facts are interactions are much more important for the datasets in GLUE. Currently, there is no experimental evidence that contradicts this hypothesis.
> > > >
> > > > If you want to claim that TNF is better at reducing the noise or modeling the word interaction in general, I suggest that you can plot a Figure like Figure 1 in [3]. Using the experiments that are already done in [3] and [1], you can compare TNF with other ways of increasing the model size (e.g., simply increasing the hidden state size or adding the memory capacity like [1]). Given the same amount of extra parameters, if TNF boosts the performance quicker (with a larger slope) than other approaches, I will agree that some of the improvement might come from reducing the noise. In the end, if you find that your slope is not significantly better than other ways of increasing the model size, I strongly recommend you to rewrite the paper to tell a different story (e.g., the hypothesis I mention above and the potential application for the case of handling mismatch between the training corpus and downstream tasks).
> > > >
> > > > [1] Lample, Guillaume, et al. "Large memory layers with product keys." Advances in Neural Information Processing Systems. 2019.
> > > >
> > > > [2] Petroni, Fabio, et al. "Language Models as Knowledge Bases?." Proceedings of the 2019 Conference on Empirical Methods in Natural Language Processing and the 9th International Joint Conference on Natural Language Processing (EMNLP-IJCNLP). 2019.
> > > >
> > > > [3] Kaplan, Jared, et al. "Scaling laws for neural language models." arXiv preprint arXiv:2001.08361 (2020).

---

> > > > > ### Author Response · Authors · 2020-11-22
> > > > > **Third Reply to AnonReviewer1**
> > > > >
> > > > > Thanks for your clarification! We are running the 'BERT expanding hidden states' experiment as you suggested to see which explanation is more true. Since probably it cannot run to an okay stage before the rebuttal ddl, we try to find more clues to tell us which explanation is more true in our existing results on BERT-Large. Because BERT-Large has larger parameter size than BERT (345M vs. 110M) and is both deeper (24 layer vs 12 layer) and wider (1024 hidden states vs 768 hidden states) than BERT.
> > > > >
> > > > > In Appendix, we show that TNF can achieve even larger improvements on BERT-Large (1.2 points compared with 0.8 points on BERT). Recent studies show that larger models have smaller margins for improvements when their model size is further increased or even doubled[1][2]. See Figure 2, 3, 4 and 5 in [1] and Figure 4 (left) in [2].
> > > > > If TNF's performance improvement is purely brought by the increased parameter size, then according to the references above, it is likely that BERT-Large-TNF's performance - BERT-large's performance < BERT-base-TNF's performance - BERT-base's performance.
> > > > > However, we observe from the experiment that TNF can achieve even larger performance improvement on BERT-Large compared with BERT-Base. We think it may suggest that the increased model size is not the only reason for TNF's performance gain. Please let us know your further comments and suggestions.
> > > > >
> > > > >
> > > > > [1] Li, Zhuohan, et al. "Train large, then compress: Rethinking model size for efficient training and inference of transformers.", (ICML 2020).
> > > > >
> > > > > [2] Kaplan, Jared, et al. "Scaling laws for neural language models." arXiv preprint arXiv:2001.08361 (2020).

---

### Official Review · AnonReviewer2 · 2020-11-02
**An interesting way to accelerate pretraining, it would help to add more analyses and details.**

**Rating:** 6
**Confidence:** 4

**Review:**

This work aims at accelerating pre-training by leveraging the contextual embeddings for the rare words. It is argued that the inadequate training of rare words slows down the pre-training. The authors then proposed to keep a moving average of the contextual embeddings for the rare words and use it to augment the input embeddings of the rare words. This technique is applied to BERT and ELECTRA and is shown to improve over the baseline.

Strength:

1. This work proposes a simple approach to accelerate the pre-training, with only a small memory and compute cost during training. The empirical study on BERT and ELECTRA supports the claimed improvements.

2. It provides an interesting view towards the rare words problem that the rare word not only has worse embeddings but also slows down training of the whole model.

Weakness:

1. It is argued that the proposed approach helps with rare words problem. But it will help to add more experiments to see how much more benefit we can get from it. For example, maybe the use of contextual embeddings are actually helpful for all the words or sub-words instead of just the rare words.

Specifically, regarding " we define keys as those words with occurrences between 100 and 500 in the data corpus", How are the range 100 to 500 chosen? Have you tried it on words appearing lower than 100 or higher than 500? As mentioned above, it would be interesting to see if this approach can be applied to more words or subwords to get even more gains.

2. Some design choices needs more details or explanations.

For example, why does the NoteDictionary use "words" instead of "sub-words" as keys? It seems using "sub-words" could cover a broader range of sentences with a NoteDictionary of the same size. It will also be easier to use during pre-training, for example, you could use the contextual embeddings to improve the word embeddings of the sub-words directly to avoid having an extra NoteDictionary.

Another example is how the window size is chosen, since it seems an important new hyperparameter.

---

> ### Author Response · Authors · 2020-11-17
> **Response to AnonReviewer2**
>
> We thank Reviewer 2 for appreciating our work and providing insightful comments. We appreciate Reviewer 2's highlight of our view that the poor embeddings of rare words could slow down the training of all model parameters. We find that you have concerns about several design choices of our work. We try to address them below.
>
> * Regarding using words or sub-words as keys in the note dictionary
>
> This is a good question. Sub-word tokens are indeed easier to be applied as keys for the note dictionary and could potentially cover more of the corpora. However, we find that quite a few rare sub-word tokens, either generated by BPE or in Google's word piece, don't have specific understandable semantic meanings to humans. Here we list a few ''195@@, ⅓, canter, elids, al, ch,  di'', separated by commas.
>
> For such sub-words, their contexts can be very diverse due to their vague semantic meanings. Therefore, saving notes for those sub-words would potentially bring irrelevant context into the current sentence and even add noise into the learning process, which we found not improving the pre-training. While if we save notes for words instead of sub-words, given that most words have concrete semantics, their notes can act as effective auxiliary semantics to enhance the current input sentence.
>
> * Regarding using different occurrence range other than (100,500)
>
> Thanks for this question. We have tuned this frequency range to build the note dictionary before the submission. We apologize for not having put the results into the paper. Specifically, we have tried setting the range to be 50-100 and 100-1000.
>
>
> Range 50-100 performs almost identical to the BERT baseline. The reason is that rare words with numbers of occurrences between 50-100 can only cover 0.3% of the whole training corpora. With such low coverage, very limited training sentences would be enhanced by rare word notes. As a comparison, words whose occurrences between 100-500 cover 2% of the training corpora, which is shown that could impact the pre-training empirically. We cannot further include lower-frequency (e.g., < 50) words in the dictionary as the number of such words would exponentially increase due to the long-tail problem of language corpora, occupying a massive amount of GPU memory.
>
> Range 100-1000 performs similar to 100-500, which suggests the rare word range in TNF is robust to some extent. Although TNF targets rare words, we do agree with the reviewer that it is worth trying to apply TNF to common words (or even all words) and check the performance. However, we may not be able to finish the experiment within the rebuttal period (due to the long pre-training and hyper-parameter tuning time). We will explore it as future work.
>
> * Regarding using different window size
>
>
> Window size is a hyper-parameter in our method. We have provided the ablation study for this hyper-parameter (and other hyper-parameters) in the appendix. See Page 12. The experimental results show that a larger window size usually leads to better performances.
>
> ------
> We hope the above responses can address your questions about the design choices. Please let us know if you have further questions!

---

### Author Response · Authors · 2020-11-24
**General response**

We thank AC for handling this paper and thank all reviewers for their kind help and useful suggestions. The comments have enlightened us to think deeper and made our work more solid than before.

We will add those discussions and new experimental results into the paper, including:

1. The experiments on sentences without rare words which demonstrate that the Transformer model is improved by TNF.

2. Make a better presentation regarding the reasons behind why TNF works.

3. Add discussions about the empirical comparison between using word2vec as notes  v.s. using BERT output as notes to show that using BERT contextual embedding does improve the performance.


Due to the intensive discussions and conducting multiple experiments during the rebuttal, we can hardly finish the revision of pdf before the rebuttal ddl. We will update the pdf as soon as possible.

Thanks!

Paper416 Authors

---

### Decision · Program_Chairs · 2021-01-07
**Final Decision**

**Decision:**

Accept (Poster)

**Comment:**

The authors propose an approach for pre-training that involves "taking notes on the fly" for rare words. The paper stirred a lively discussion on the reasons for the reported results, which the authors followed-up with new experiments and findings that convinced the reviewers that indeed their approach is valid and interesting. Thus, I am recommending acceptance.